# Toward Point-of-Interest Recommendation Systems: A Critical Review on Deep-Learning Approaches

Sadaf Safavi [1], Mehrdad Jalali [1,2,*] and Mahboobeh Houshmand [1]

1 Department of Computer Engineering, Mashhad Branch, Islamic Azad University, Mashhad 9187147578, Iran; safavi@mshdiau.ac.ir (S.S.); houshmand@mshdiau.ac.ir (M.H.)

2 Institute of Functional Interfaces (IFG), Karlsruhe Institute of Technology (KIT), Hermann-Von Helmholtz 4 Platz 1, 76344 Eggenstein-Leopoldshafen, Germany

* Correspondence: mehrdad.jalali@kit.edu

**Abstract:** In recent years, location-based social networks (LBSNs) that allow members to share their location and provide related services, and point-of-interest (POIs) recommendations which suggest attractive places to visit, have become noteworthy and useful for users, research areas, industries, and advertising companies. The POI recommendation system combines different information sources and creates numerous research challenges and questions. New research in this field utilizes deep-learning techniques as a solution to the issues because it has the ability to represent the nonlinear relationship between users and items more effectively than other methods. Despite all the obvious improvements that have been made recently, this field still does not have an updated and integrated view of the types of methods, their limitations, features, and future prospects. This paper provides a systematic review focusing on recent research on this topic. First, this approach prepares an overall view of the types of recommendation methods, their challenges, and the various influencing factors that can improve model performance in POI recommendations, then it reviews the traditional machine-learning methods and deep-learning techniques employed in the POI recommendation and analyzes their strengths and weaknesses. The recently proposed models are categorized according to the method used, the dataset, and the evaluation metrics. It found that these articles give priority to accuracy in comparison with other dimensions of quality. Finally, this approach introduces the research trends and future orientations, and it realizes that POI recommender systems based on deep learning are a promising future work.

**Keywords:** point-of-interest recommendation systems; deep learning; machine learning; systematic review

## 1. Introduction

Today, due to the developing infrastructure of the internet networks, smartphones, and accessing the majority of users, the large datasets of information and choices are organized for them, and finding related and adequate information is complicated for users. Users expect personal content to be available in novel e-commerce, entertainment, and social networking systems. With the addition of location to social networks, location-based social networks (LBSNs) were introduced. Indeed, these networks function as a bridge between reality and online social networks. Using Global Positioning System (GPS) information on mobile phones, and analysis of user movements to suggest point-of-interest (POI), there is no indication obtained except the latitude and longitude in the user history, and there is no way to find out if the coordinates are related to a coffee shop or cinema.

With expanding services and LBSNs such as Yelp, Gowalla, Foursquare, Brightkite, the feasibility of study about personal offers has been prepared. Users can easily check-in their physical life experiences and share them in diverse geographical spots [1]. These networks keep a large database of POIs and let users check-in their present spot with their smartphone and display it via a POI. People save their situations and paths and propagate

them on the channel of the internet. With the creation of this technology, users share their local data such as images or notes which have locative labels on social networks and transmit their experience to friends, and this matter can provide an opportunity to find new friends. The structure of these networks can be split by the user, geographical location, and POI content levels.

The outcome of the performance of location-based social networking services is the acquisition of large datasets from which location history, the structure of social relationships, mobility behavior, and user characteristics can be extracted. These rich sources of information have a high ability to recognize users. Because a client's position history in the real reflects his penchant and behavior. Therefore, persons with similar location histories have more potential to share common preferences and behaviors. Computing similarity between clients leads to the requirement of innovative tasks and systems—one of the most significant of which is the recommendation systems that analyze user behavior and requirements—to offer the most proper items and valuable information. This system is a technique that is proposed to deal with the issues caused by the huge and growing volume of information and plays a key role in obtaining job information, decision process analysis, location-based advertising, and increasing industry revenue. Thus, in the past few years, the location-based recommender systems for users, the market, and for academic research have attracted much attention, and they also helps advertising companies to plan, activate, and analyze successful marketing campaigns. The main purpose of recommender systems is personal recommendations, user satisfaction, and establishing long-term relationships with users. Conventional recommender systems, suggest POIs to users that they do not appreciate. Hence, the need for a method to recommend a POI with high accuracy is essential.

In recent years, employing deep-learning strategies has seen a persistent rise in artificial intelligence works such as POI recommendation, computer vision, and natural language processing where main traits can be exploited extremely and successfully [2–8]. Universities and industries use this technique for a wider range of applications due to its ability to solve many complex tasks while delivering acceptable outcomes because deep learning outlines a representation-learning method that is dependable for learning data representations with several uncomplicated components. Every component analyzes the input of high-level representations from the previous module (from the low-level feature extractor module) [9,10]. Thus, dissimilar deep-learning novels gain very good outcomes to extract the essential high-level features that can be proper for recommendation services. Recent advances in recommender systems based on deep learning, by overcoming the challenges of traditional techniques and gaining great recommended quality, have obtained applicable importance.

### 1.1. Problem Statement

The difference between this study and the other previous work is that there are very few systematic reviews on the employment of deep learning in the POI recommendation field and on studying the advantages and disadvantages of their methods that can introduce recent state-of-the-art approaches and current developments. Although some methods have studied traditional recommender systems [11–16] and a variety of recommender systems based on deep-learning techniques [17,18], they have not been reviewed characteristics in the POI field with up-to-date and novel state-of-the-art features. Due to the increased interest in the POI recommendation field, this survey aims to provide an overview of recent approaches to POI recommendation systems based on deep-learning techniques and categorizes advanced methods of deep learning. The existing constraints are highlighted, and new directions will help make future decisions and be the starting point for novel research in this domain. This work categorizes state-of-the-art studies based on the deep-learning method adopted, the dataset used, and the evaluation metric, and it observed that most of the selected research considers accuracy as a quality dimension. It also discusses the

features and influential factors in POI recommender systems and represents the traditional machine-learning methods and their strengths and weaknesses.

### 1.2. Definition of Research Question

In this work, our research questions are the following:

RQ1. What are the types of POI recommender systems, influencing factors, and challenges in their recommendation?

RQ2. What are the traditional machine-learning methods and deep-learning models that have been considered in recent POI recommender systems?

RQ3. What are the most widely employed evaluation metrics and popular datasets to evaluate POI recommendations based on deep learning?

RQ4. What are the most significant future research trends and open issues?

### 1.3. Contributions of This Survey

The purpose of this systematic literature review is to specify the recent state of the art in the field of deep-learning-based POI recommendations. This work can contribute to the success of research areas in universities and industrial centers, and researchers with rich knowledge of the factors influencing POI recommender systems and traditional machine-learning approaches can select appropriate deep neural networks and combine different methods for solving their proposed tasks. Authors can consider the strengths and weaknesses of methods and review evaluation criteria and popular datasets. This paper supplies an overview of the state of the art and recognizes the recent trends and future directions in this research domain.

The structure of this paper is represented as follows. Types of recommendation systems, influencing factors, and challenges in POI recommendations are outlined in Section 2. In Section 3, the traditional machine-learning approaches and the latest deep-learning methods in POI recommendation systems are described; this article also discusses their strengths and weaknesses, and it presented the classification framework about the recent state of the art in this field. Section 4 categorizes the popular datasets and evaluation metrics of the latest state-of-the-art features to evaluate POI recommendations. This systematic review concludes in Section 5 with the most influential future research trends and open problems.

## 2. Overview of POI Recommender Systems

In this part, we first provide the recommender systems and the types of recommendation techniques, then the POI recommendation and factors influencing these systems are outlined. The significant challenges facing POI recommendations are also discussed. This section tries to answer the first question of the survey.

### 2.1. Recommender System

The recommender system is an intelligent system that provides personalized recommendations which achieve user satisfaction by extracting and filtering information from big data based on the user requirements and preferences [19,20]. A recommender system consists of three steps:

- Acquire user preferences based on explicit and implicit data;
- Make recommendations using appropriate techniques;
- Present recommendation results to users.

The recommender system tries to make recommendations that clients may have an interest in. [21]. Generally, recommended lists are based on item features, user's preferences, user former interactions with the item, and other further information such as temporal data (such as sequential recommendations) [22] and location (such as POI recommendation) [23], and social data [24].

The recommender systems have become more favored in the last few years due to their diverse applications and have been highly regarded by the university and industry;

however, they still have a variety of challenges, including dynamics and changes in communication, time, location, and users' requirements. Another problem is the cold start issue; the cold start issue is categorized into user cold start problem, item cold start. One of the other problems is the existence of large datasets, and the scoring matrix is too sparse and the problem of data sparsity occurs. Security and reliability are other challenges [25].

However, a quality recommender system should simultaneously consider factors of user preferences [26,27], time stamps [28,29], check-in correlation, social network comments, and friend importance [30,31]. The following are the types of recommendation methods in recommender systems that were discussed in most approaches.

### 2.1.1. Recommender Systems Based on Collaborative Filtering

The recommendation system provides suggestions based on the client's past behaviors and their similarity to the past behavior of other users [32,33]. Prior behavior can be in the form of explicit feedback such as user scores and comments that it evaluates according to its interests, or in the form of implicit feedback that the system performs the required analysis according to the user's search, viewing, and purchase records. Collaborative filtering is among the most popular algorithms in recommendation systems. This technique allows the client to make decisions rapidly and easily [34] and can also select a product according to consumer preferences among a large number of candidates [35–40].

The CFSKW model [41] proposes a POI recommendation technique that utilizes geographical influence, and the role of geography in the recommendation is mentioned separately. The quality of the recommendation is improved by integrating collaborative filtering and a method for spatial kernel weighting. The two important factors that are tuned to the CFSKW model are the kernel bandwidth and the coefficient of incorporating user preference with geographical impact. The suggested method experimented on New York and Tokyo Foursquare datasets. According to the findings of this study, low bandwidth as well as medium to high geographical coefficient yield more accurate results. A proposed dynamic bandwidth-based algorithm has shown promising outcomes at disproportionate densities of POIs, such as the Tokyo dataset. The recommended method has been evaluated with state-of-the-art approaches and analyzed with Precision @ N and Recall @ N metrics. This evaluation indicates that the CFSKW method in terms of Precision@5 and Recall@5, respectively, at 3% and 5.1% in the New York dataset and at 1.1% and 1.2% in the Tokyo dataset performed, better than the reference algorithms, and the use of the dynamic bandwidth geographic similarity technique has been effective in improving performance.

### 2.1.2. Recommender Systems Based on Content Filtering

The recommender system can suggest a new item by searching for item features [42–44], but in collaborating filtering systems, to make a recommendation, a score item is required first. Content-based recommender systems can provide detailed tips using active user profile information (such as purchase history, queries, ratings) even in a situation where collaborative filtering systems have new items and sparsity challenges [45,46].

In this paper [47], a novel recommendation algorithm in the field of IoT and remote device control with the help of smartphones is proposed, which is a combination of content-based and collaborative filtering methods and uses context information such as orientation and location. Other tasks considered in this work are location detection, ambiguity handling, recommendation making, orientation detection, item extraction, and profile creation. This proposed method executes better in terms of the level of personalization due to the focus on the role of user orientation. Future studies can be applied to location-based recommendations such as restaurants and rest stops. Increasing user privacy is also a very important issue.

### 2.1.3. Recommender Systems Based on Other Methods

Recommender Systems Based on Graph

These systems employ a graph method where items and users such as nodes and lines are the transactions between user–user and user–item. The recommender approach based on a graph involves the construction of a graph that represents data, and with graph analysis, recommendations can be made [23,48]. This method can realize the similarity of items and also evaluate which users do score or purchase with them. The benefit of this recommendation system is that after the creation of the graph, you can easily add a node and create a connection between the nodes [49,50]. Of course, checking new items in a graph is no easy task.

Recommender Systems Based on Knowledge

This technique is based on user interest and does not return to prior user preferences. It is used to make knowledge-based recommendations of the client and the products, and it makes a difference to other techniques [51,52]. Surely, it is quite difficult to create and then keep updated the knowledge base, because it needs enough skill and expertise for the intended issue. Knowledge-based recommender systems are beneficial in scopes where rating systems (collaborative filtering and content-based filtering) do not operate [53].

They are effective in recommending certain items such as luxury goods, automobiles, and real estate which are not usually bought. In these instances, techniques based on knowledge have proven successful. These systems utilize the item properties and then create a user profile for recommendations to users.

Recommender Systems Based on Demographic

These systems use demographic features for classification that can generate demographic information for ranking [54]. A recommender system categorizes user information based on their characteristics such as country, status, name, age, gender, and work [55]. The benefits of the demographic filtering method are that no user rating history is required and it makes recommendations quickly and easily [56]. However, due to security problems, it is not easy to acquire demographic information. Generally, demographic filtering methods alone may not provide the best suggestion. These systems' accuracy is improved when incorporated with other methods such as techniques based on knowledge [57]. Another constraint is the lack of sharing personal information by users for the fright of misuse in online networks.

Hybrid Recommender Systems

To improve recommendations, this system is synthetic to some methods. The best properties of every technique such as collaborative filtering and content-based filtering recommendations have been incorporated to recommend accurately, and its problems will be solved, and their predictions will be employed in recommendations [58,59]. The authors of the article [60] merged the collaborative filtering techniques with deep neural networks to gain the ability to learn features and user items. The mixture of methods makes it possible to improve the efficiency of the system.

The paper [61] suggested an algorithm for online marketing recommendations that integrate content and collaborative filtering. This fusion recommendation technique solves the new client problems by relying upon content filtering, sparsity of data based on collaborative filtering, and cold start challenge problems. The interests of existing users are first discovered. Then the potential interest of the client is extracted from the model of combined similarity of content and behavior by considering the similar user group of the target user and predicting the interest of the user for feature words. After that, the existing and potential interests of the user are combined. Finally, to provide appropriate recommendations, the similarity between the content and the fusion technique is estimated and then clustered via K-means. This article has been evaluated for the MovieLens data set

by recall, precision, and hybrid similarity metrics. The proposed hybrid method can solve the mentioned challenges and has a great effect in terms of recall, accuracy, and diversity.

### 2.2. Social Recommender Systems

Integration of social networks and recommender systems has great advantages. Millions of active clients dedicate a portion of their time to social networks. Social activities include creating a user account, communicating with friends, joining some communities, making a comment, movie, or image, tagging resources, and doing ratings [62]. This data can be used to improve the predictive efficiency of recommendation systems such as friend recommendations and social relations, and it can reduce the issue of information overload [63]. The offer of recommender systems is more based on friends' ratings, not anonymous users. Content information such as time, status, situation, and mood of the user is the most important factor in proper recommendations. Location is also a very important and vital factor for the user; it has a great impact on suggestions based on the preferences of the user [64–66].

### 2.3. Point-of-Interest Recommender Systems

One of the research scopes for recommender systems is point-of-interest recommendation. POI means any place that the user can visit (museum, park, cinema, art gallery, restaurant, coffee shop, shopping center, etc.). With the development of GPS and mobile technologies, clients' use of location-based applications has increased significantly. A location-based social network (LBSN) is a social network whose location is one of its dimensions and operates as a gateway between users and places, between clients and between places [67]. Users can relate with their friends on these networks, checking in at their POI places (such as restaurants, tourist spots, shopping places, etc.) and sharing them at a specific time and date, write tips, upload their images, share comments, and as a result, much information about offline physical activity is provided online. Foursquare, Gowalla, Yelp, Facebook, WeChat, Twitter, Instagram, etc. are the sample of services and location-based social networks. LBSNs use this rich information (social communications, check-in history, coordinates, and order of POIs) to integrate user preferences on locations and on recommender systems (location recommendation, friend recommendation), and location-based services (event-based suggestion, urban planning, marketing decisions, mobile-based viral marketing, etc.) are used. In addition, it always provides a new perspective through which urban structures and related socio-economic performance can be demonstrated, and road networks and the popularity of POIs can be estimated. Urban flows can be analyzed in the urban environment. Major urban/emergency events can be identified. Major urban/emergency events can be determined, and the social and economic impacts of cultural investors are identified [68].

POI recommender systems can employ types of entities in LBSNs, including the following:

- *User*: individuals who are members of LBSNs to check-in;
- Location: the places that are visited by users;
- *Region*: an important factor in LBSN that can distinguish these systems from its traditional sample. This entity has two features: latitude and longitude. Due to this entity, users are more intent on visiting POIs that are close to their current location or that are placed in their area of interest;
- *Time*: This is very important for location-based recommender systems, and users may experience different behaviors at different times and events;
- *Social relations*: Some users are friends with each other in this network, which is called social relations. This feature is utilized to increase accuracy in location-based recommender systems because friends on LBSNs have more common interests than others.

Figures 1 and 2 are constructed using the VOSviewer software tool, showing a network visualization for the supplied keywords presented in the Scopus database articles by 2022. More specifically, the figures show the co-occurrence network and the topic clusters for the POI recommendation keywords and their connections.

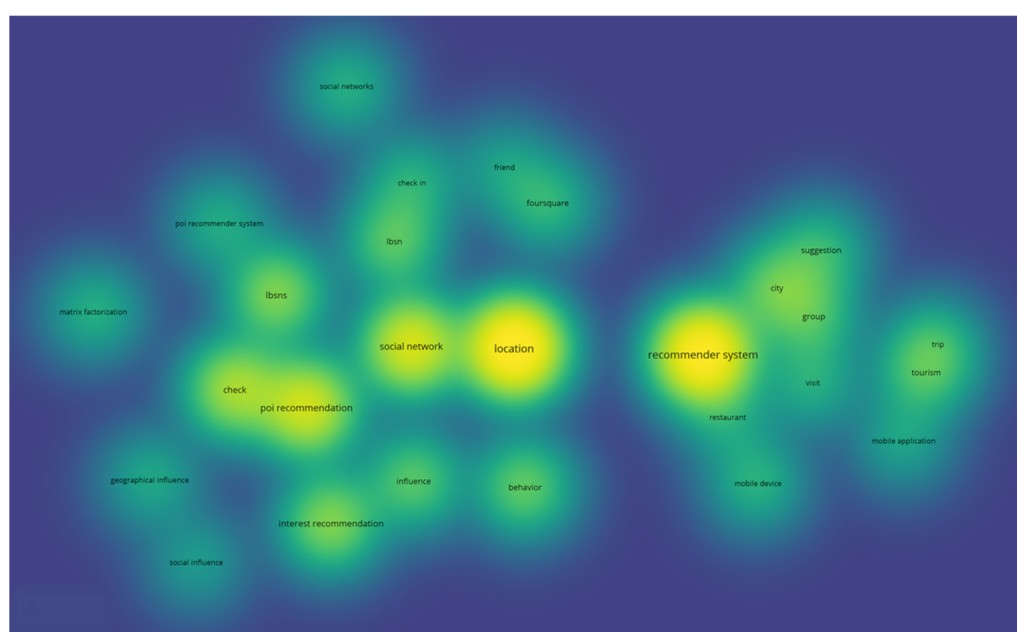

**Figure 1.** Network visualization for the POI recommendation keywords with VOSviewer.

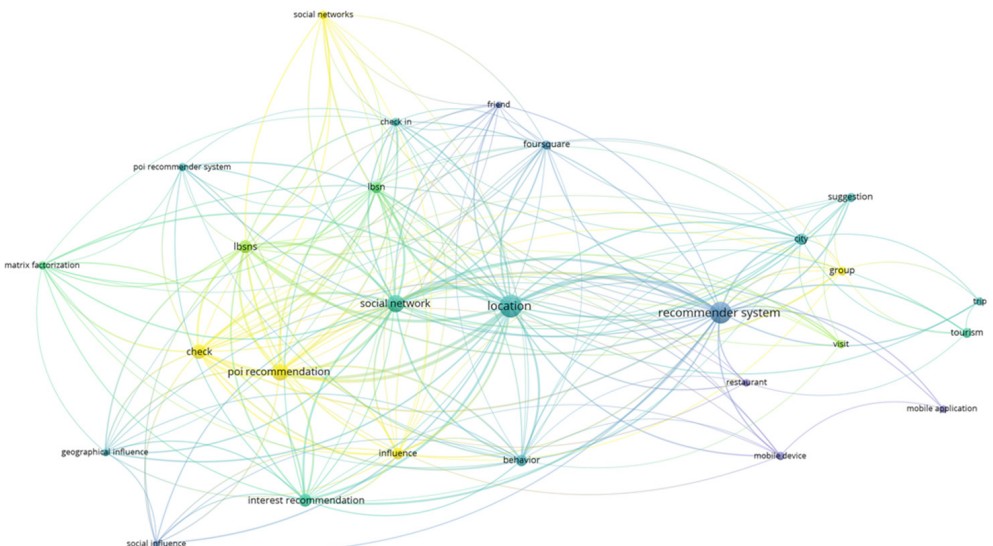

**Figure 2.** Network connections for the POI recommendation keywords with VOSviewer.

Therefore, the existence of an environment that can integrate all these features of the entities according to their different types can be effective in improving the accuracy of the recommender system [3,6,69].

### 2.4. Factors Influencing POI Recommendations

Due to spatial and temporal characteristics derived from physical limitations and heterogeneous data such as geographic location data and user descriptions about the place, check-in action is a complex intention of various factors. Research in the POI recommendation field according to the factors influencing the user's check-in activity is categorized and reviewed. Figure 3 illustrates the POI recommender system and the factors influencing it.

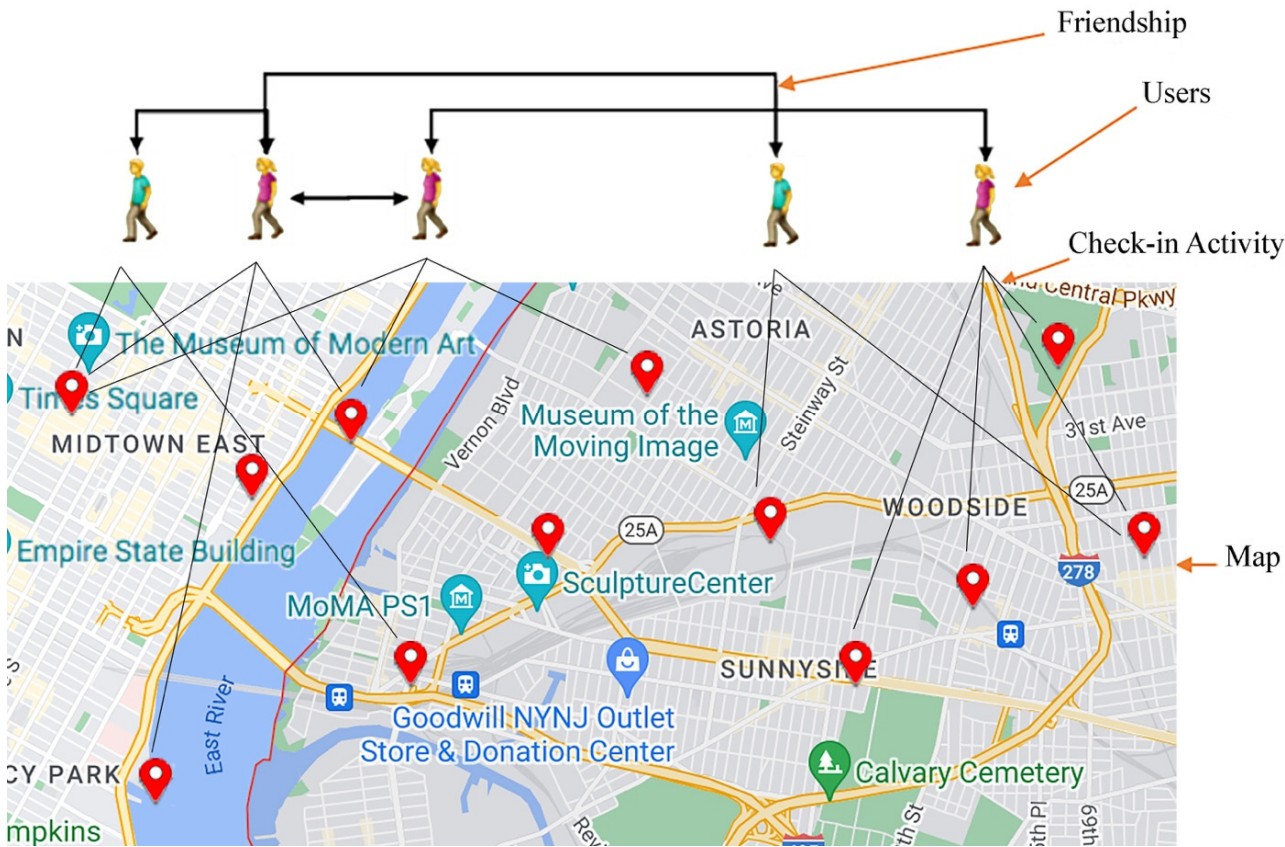

**Figure 3.** Demonstration of the POI recommender system and the factors influencing it.

In general, the factors that influence the POI's recommendation can be classified as follows:

- *Geographical influence*: It is also called spatial influence. The behavior of the user's check-in depends on the geographic characteristics of the location and has a tremendous impact on users' visiting behaviors; the most important feature of POI recommender systems is compared to traditional ones [12,70,71]. According to Tobler's first rule [72], users prefer to visit locations close to them instead of places far away, and as the distance increases, the probability of visiting a new location decreases.
- *Temporal influence*: For POI recommendation, temporal influence is a critical factor because check-in takes place with a certain pattern due to physical limitations [73]. As to routines in our daily lives, there are various possibilities for different places at different times of the day on different weekdays. Users' check-in behaviors occur on weekdays due to work, lunch, or in the evening, and the POIs they check-in are next to work or home. On weekends, the behavior of check-ins changes, and that of users changes over time [74–77].
- *Sequence influence*: One of the factors that affects the check-in of a user's behavior is the order of the check-ins. The destination of the client may have been influenced by the previous POIs he has visited [78–80].
- *Social influence and importance of friends' behavior*: The assumption is that the clients in their decisions are persuaded by the explanations and suggestions of their friends compared to other users. The importance of a friend [3] evaluates the influence of a friend while visiting the POI. Numerous studies [81–86] indicate that social relations are beneficial for the recommender systems, and the use of social factors to reinforce traditional recommendation systems has been investigated, both in memory-based methods [87,88] and in model-based techniques [89–91]. Attention to social

influence and friend impression in POI selection has improved the recommendations of traditional recommender systems.

*2.5. Challenges of POI Recommender Systems*

In addition to the challenges listed in Section 2.1, the following describes some of the issues with these systems versus traditional recommender systems:

- *Heterogeneous data*: LBSNs involve different kinds of information, containing not only geographical data about the position, locate descriptors, and check-in history, but, furthermore, media knowledge (e.g., comments and tweets on customers) and information about clients' social relationships. This heterogeneous data illustrates user movements from different points of view, illuminating POI recommendation systems with various methods [12,31]. Extensive scientific research indicates that the social relationship between clients is a significant segment of the POI recommendation. In [27], a hybrid random walking method based on a graph with a star pattern was suggested, and it integrated multiple structures of heterogeneous links. In this article, frequency or the rating of social check-in is considered as an effective score to recommend;

- *Physical limitations*: Compared to services such as watching a film on Netflix and buying online from Amazon, physical restrictions limit check-in activity. Such constraints create check-in activity in LBSN, showing significant temporal and spatial features. For example, stores normally prepare tasks in a finite time;

- *Complex relations*: To provide online social networking tasks such as Instagram, Facebook, and Twitter, location is a new feature that creates novel relationships between spots as well as between spots and clients. Further, the spots where activities are shared change customer relationships who are ready for new friendships with geographic neighbors and to impact each other psychologically. Geographical proximity has a significant impact on the check-in of user behaviors and on points of interest. At LBSNs, customers are interacting with POIs physically, a unique phenomenon that is distinguished from traditional suggestions. Article authors [92] proposed a kernel density estimation approach for personal location recommendation based on social and geographic features (iGSLR) and used this structure to plot personal geographic and social influence. They estimated the interval distribution among each pair of places by kernel density estimation;

- *Data Sparsity*: The major issue in the POI recommendation strategy is data sparsity. Once someone is in a place and checks in a spot, the place and the time are registered by a check-in label and suggested to other clients to visit this place. For each person, repeating a visit to different locations is an item in the user-location sheet (matrix). Because not all places are visited by all users, a significant sparsity can be seen inside the matrix;

- *Rich context*: Different context knowledge of LBSNs, such as social relations, nearness to users, and geographical coordinates of the POI, can be observable. Context knowledge of this network is obscure and inadequate, making it difficult for the POI recommendation. For instance, the POI geographic distance of the concerned place can totally affect the user trajectory and behavior. Sometimes users visit a place such as a cinema that is close to their work or home and then suggest this certain place to friends.

## 3. Traditional Machine-Learning Methods and Deep-Learning Models in POI Recommendation

In this section, the second question of our article is answered. Traditional machine-learning methods are offered for recommendation in POI recommender systems. Then, deep-learning methods are introduced for more detailed recommendations in these systems, and the significance of their utilization in the field of POI recommendation is discussed. Recent state-of-the-art features are expressed in these areas. Figures 4 and 5 are showing

the network visualization and their connections for the supplied keywords presented in the Scopus database articles by 2022. More specifically, the figures show the co-occurrence network and the topic clusters for the POI recommendation with deep-learning keywords. Finally, the advantages and disadvantages of these methods are reviewed.

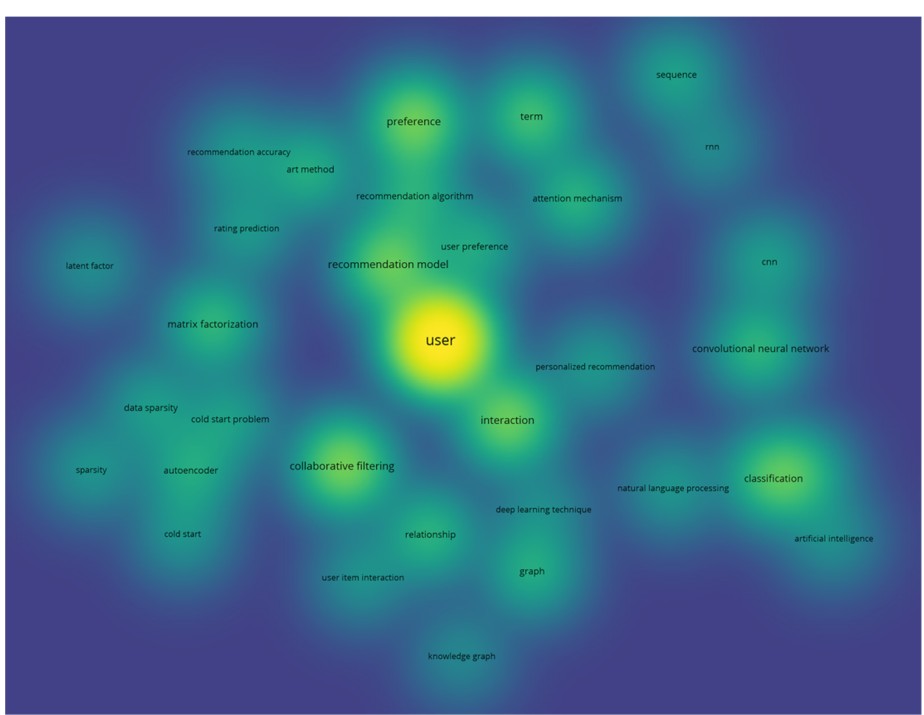

**Figure 4.** Network visualization for the POI recommendation with deep-learning keywords with VOSviewer.

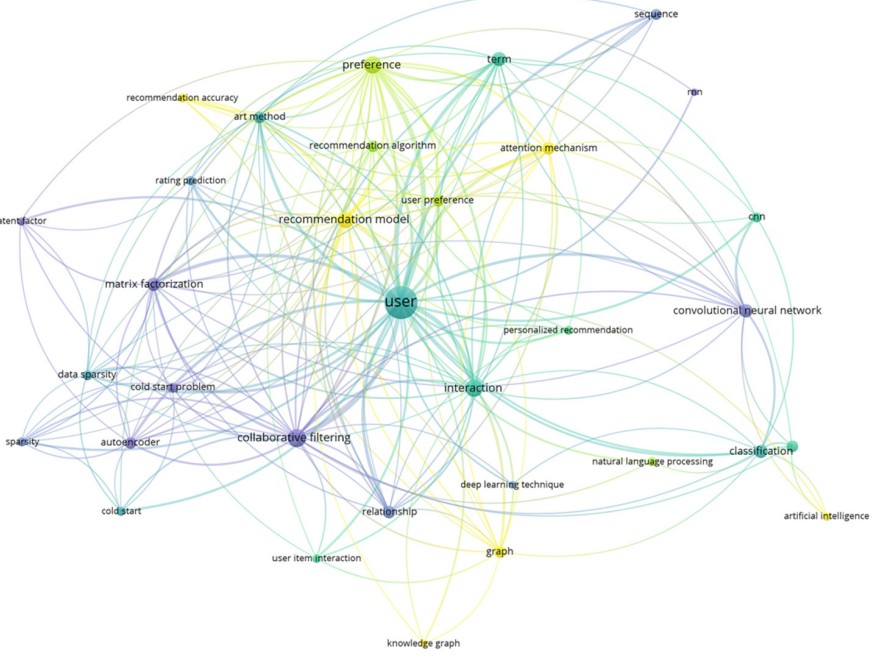

**Figure 5.** Network connections for the POI recommendation with deep-learning keywords with VOSviewer.

### *3.1. POI Recommender Systems Based on Traditional Machine-Learning Methods*

Machine-learning is one of the artificial intelligence subsets, and it is a general name of models that gain information and deduce data. The main centralization of machine-learning is for developing programs that will be able to access data and use it for their-learning and generate output due to their instruction. Traditional machine-learning algorithms have problems understanding complicated relations in Big Data because they are not designed for big data processing. Based on the issues of POI recommendation systems, it is evident that the development of these systems has been outdated and complicated with high accuracy and limitations such as tunning and heavy calculations using traditional machine-learning algorithms. However, this model is still employed in some deep-learning models.

### 3.1.1. POI Recommendation Systems Based on a Collaborative Filtering Method

Collaborating Filtering is a technique that has often been used for recommender systems [93–95]. The essential procedure is that if users have similar preferences about items, they probably display similar preferences about other items. It was analyzed in two ways: memory-based and model-based. In memory-based methods, similarity rates are calculated using the value of user ratings, etc. In the model-based approach, which is used in POI recommender systems, the model is trained from the prior ratings by employing machine-learning algorithms, and further predictions are made. This approach is proposed by the user and is item-based. The main goal of collaborative filtering user-based is to find similar users for the target user [96,97]. Several methods have been used to estimate the similarity between users. Choosing the correct similarity function is an important step for the recommender system. The two similarity functions which are used more are Pearson Correlation Coefficient and Cosine Similarity. Item-based collaborative filtering [98,99] computes two items that are similar to one another [100,101].

Generally, there are two approaches to implementing collaborative-filtering-based recommendation systems: nearest neighborhood analysis [102,103] and latent factor analysis [104,105]. In the first method, the recommender can compute the connection weight of the users or items to choose the nearest neighbors. The user's potential preferences are then predicted by examining the user's preferences for a new item based on last user data or highly related items, but the latent-factor-analysis-based method is a low-dimensional representation of items and users by which their dependencies can be accurately modeled. This method is derived from matrix factorization (MF) [106] approaches, and it can discover low-rank feature matrices to define data, and user ratings are factorized to an item and user feature vectors. This creates a series of loss functions based on known target matrix data that delays the desired latent factors, and then it minimizes the resulting loss functions with respect to the desired latent factors to a technique that provides a low ranking with acceptable representation. This method focuses only on known data and is efficient in addressing the sparse issue of recommendation [107–110].

Q. Yuan et al. [111] offered a time-aware POI recommender system; when calculating the users' similarity in collaborative filtering, the time will be added and the time-aware effect in previous data will be considered. PR-RCUC [26] model proposes a novel POI recommendation technique that integrates the region-based collaborative filtering method with a user-based mobile context. A challenge in this work is data sparsity, and also, it is difficult to provide a logical explanation to the user in relation to the suggestion and visit to the desired location. To reduce the data sparsity problem, this model first clusters spots in various regions and combines the region factor with the collaborative filtering method. The next task of this technique is to create the mobile context for a client such as geographical distance and categories of the location. Finally, it fuses the above two parts and presents the PR-RCUC method. In this model, two datasets from Foursquare are examined and also employ three widely used criteria—accuracy, recall and F1-score—to estimate the proposed model. Experimental results indicate that the PR-RCUC algorithm outperforms some famous recommendation methods. In the continuation of this work,

different geographical clustering methods can be examined, and more contexts such as season or weather can be considered.

### 3.1.2. POI Recommendation Systems Based on a Markov Chain Method

A stochastic model outlines the sequence of probable events where the possibility of each event is dependent only on the status achieved in the prior event. Therefore, the probability of events happening in such a model relies only on the prior time, and different events do not interfere with the probability [112,113]. Essentially, there are two types of models of Markov chain: continuous-time Markov chain (CTMC) [114] and discrete-time Markov chain (DTMC) [115]. In CTMC, processes are continually happening over time. In DTMC, the time parameter is discrete, and a Markov chain will be created with the determination of a random data sequence. DTMC-based approaches are most commonly used in research on POI recommender systems.

The authors [116] proposed a new sequential prediction model according to the Markov chain model named SONG. According to this work, clients are interested in visiting past POIs in the short term and tend to visit novel POIs in the long run. This approach models the behavior and geographical impact of clients with a variable-order additive Markov chain. The Foursquare and JiePang [117] datasets are used to test this algorithm. Recall (Rec @ k) and Normalized Discounted Cumulative Gain (NDCG @ k) are employed to validate. Experimental outcomes indicate that the suggested SONG substantially enhances the performance compared with the reference algorithms.

One of the major motivations for the frequent use of the Markov chain model is that it is a random process and, with its sequential structure, can be implemented for POI problems [113,118–120]. However, the Markov chain model has many disadvantages in POI recommender systems. Many parameters (social influence, time, user preferences, etc.) must be considered between state transitions in recommender systems, but simple Markov chains only consider transitions between independent states such as places. Furthermore, the next user check-in activity is just related to the previous check-in in the Markov Model.

### 3.1.3. POI Recommendation Systems Based on a Matrix Factorization Method

Matrix factorization is a category of collaborative filtering that is one of the most common methods employed in recommender systems [121,122]. Its fundamental principle is the main matrix decomposition process, which decomposes the client interaction matrix into items by multiplying two rectangular matrices with lower dimensions. The original matrix with two different latent spaces is represented in the simple matrix decomposition method [123–126].

A new POI recommendation model based on the spatio-temporal activity center POI (STACP) suggested by Rahmani et al. [127] considers the impact of a user's spatial and temporal features jointly. This technique, based on the matrix factorization method, is statically trained according to the time feature and forms centers of spatiotemporal activity for users, and it improves the quality of the recommendation. To evaluate the performance of this algorithm, two popular datasets, Foursquare and Gowalla, and also evaluation metrics Precision, Recall, and NDCG, have been used. Experimental outcomes illustrate that the STACP model enhances statistical performance compared to state-of-the-art algorithms and illustrates the influence of utilizing geographic and temporal information in modeling client activity centers and the significance of their joint modeling. For higher improvement of this model, more information such as comments of users and social relations can be added to the algorithm.

### 3.1.4. POI Recommendation Systems Based on a Bayesian Personalized Ranking (BPR) Method

One of the models of statistical inference is the Bayesian technique in which the parameter is assumed to be a random variable with its own distribution within the parametric space. It computes the probability of a hypothesis occurring based on previous documents and information and then a decision is made [128]. In fact, Bayesian ranking is nothing

but conditional probabilities, but a very positive feature of the Bayesian algorithm is that it can prove optimality. More precisely, if the validity of the input information to this algorithm, which is used for ranking, is 100%, it can be proved that Bayes provides the best ranking compared to other methods [129]. This method is used in POI personal recommendation [130–132].

The BPRN model was introduced by Hu et al. [133] who developed a new multi-layered neighbor-based BPR algorithm to investigate hidden information in recommending systems. The authors, based on analyzing the relationship between the item and the client, and examining several layers, determine that the item without ranking can be a desired and neighboring item for a user, and define the criteria for each layer. The items are then divided into different sets and arranged, and a personal, sorted list is specified for each client based on the model provided. Five datasets Movielens-100 k (ML-100 k), Movielens-1 m (ML-1 m), Ciao, Epinions, and Eachmovies [134] were used to test this algorithm, and also Precision, Recall, and F1 evaluation metrics were selected. The proposed method shows satisfactory results on the datasets. In the future, this approach to solving data sparsity and cold start issues could integrate multiple-layer analysis and also transfer learning.

### 3.2. Deep-Learning Techniques in POI Recommender Systems

Deep learning is a subcategory of machine learning and is defined as neural networks having more than two layers that learn several levels of data representation. Traditionally, machine-learning algorithms have relied on the representation of input data. Thus, feature engineering has been one of the key research topics for a long time, and also, feature extraction operations are based on its application type and require noteworthy human efforts. For instance, in the field of machine vision, several various types of features have been introduced and evaluated; these include histogram oriented gradients (HOG) [91], scale-invariant feature transform (SIFT) [83], and bag of words (BOW) [89]. The same conditions exist in other fields, such as natural language processing (NLP) and speech recognition. Nevertheless, deep-learning algorithms perform feature extraction in a fully automated method and allow researchers to extract features without requiring knowledge in the field of the desired domain and human input [85]. These algorithms have a layered architecture for data representation defined in the upper layers of the top-level features; the low-level features are extracted in the lower layers and are capable of much more complex abstracts than representing the data in the layer.

Deep learning is capable of effectively demonstrating nonlinear and linear relations between the user and the item [135]. It extracts complex relations from within the data, from many available data sources such as visual and textual information. Training and learning are based on the improvement of a procedure that aims to decrease error in the reconstructed output. Deep learning can receive all sources of information in the input and send it to the output by classification. This highlights the most important advantage of deep learning.

According to the POI recommendation, deep-learning techniques work better than machine-learning models because issues have big data and many features and parameters that must be considered to predict the next location. Deep-learning models of recurrent neural network methods are mostly used in the next POI recommendation because the data structure for a recommendation is sequential. Methods such as attention mechanism and sequence-learning mechanism (Seq2Seq) as well as convolutional neural network algorithms are used for next location recommendations.

### 3.2.1. POI Recommendation System Based on the Multilayer Perceptron (MLP) Method

Multilayer Perceptron (MLP) is a type of feedforward artificial neural network (ANN) [136]. This model is the primary deep neural network that includes a series of fully-connected layers and determines the nonlinear relationship between entities such as users and locations.

MLP for training utilizes a supervised learning method well-known as back-propagation. Several layers and their nonlinear activation differentiate the MLP from a linear perception.

Its purpose is to decrease the error by propagation, which regulates parameters and weights. MLPs are appropriate for classification prediction issues in which inputs are allocated to a class or tag. They are also proper for regression prediction difficulties, where an actual value quantity is predicted with respect to a range of inputs. One of the advantages of this method is that neural networks are capable of generalizing, meaning that they classify an unknown model with other known models which have similar distinctive features. This indicates that noisy and incomplete inputs are categorized because they are similar to pure and complete inputs.

In LBSNs, a dual neural network is needed because the interaction between user preferences and POI attributes is a two-way interaction. Ding et al. [137] proposed RecNet network, which focuses on common places visited due to geographical proximity and classification. In the training phase, the user who visits a POI is a positive example and considers the task of recommending the POI as a binary classification and predicts the probability of the client visiting an area. One point is assigned to each POI, and the POI with the greatest score is provided to the user accordingly. They utilize the ReLu activation function for hidden layers and employ dropout techniques to reduce the problem of overfitting. The RecNet article prioritizes user presence using a binary classification method to minimize the cross-entropy loss function.

Most articles have used hybrid methods to solve POI recommendation issues that have employed the MLP in the prediction phase [138–140].

### 3.2.2. POI Recommendation System Based on the Autoencoder Method

Autoencoders (AEs) are an unsupervised technique for learning to represent data that uses a backpropagation algorithm to equalize the output of the model with the input. In recent years, its ability to describe the features of nonlinear and complex data has been considered by researchers [141]. It has two phases: encryption and decryption submodels. The encoder is employed to teach the model via an activation function that is accountable for mapping the input to the latent space. The decoder, on the other hand, utilizes another activation function to rebuild the latent space into an approximate space. Autoencoders are mainly trying to encode the data by compressing it into lower dimensions so that their features are data-specific, which can only compress the data they are trained on significantly and are different from standard compression algorithms. Another feature is Lossy: the output of the autoencoder will not be exactly the same as the input; it will be a close but weak representation. The aim of the autoencoders is to reduce dimensions and can investigate how different the output is from the input when a new vector is introduced. In recommender systems, autoencoders can be used to learn how to represent lower dimensional features in the bottleneck layer.

Ma et al. [142] employed a stacked autoencoder (SAE) technique to improve the performance of the personalized POI recommended tasks. SAEs solve the problem of extracting complex features of input data. SAE is a neural network consisting of multiple layers of AE. The output of the prior layer of the autoencoder is employed as the input of the next layer of the autoencoder, which can obtain the complex connection between the client and the location and ensure that it is hidden. Feng et al. [143] suggested a similar idea. The SDAE model used this technique to reconstruct user check-in information and POIs and obtain their best initial value, and it effectively improved the learning efficiency and performance of the matrix. The SDAE decomposition process is DAE-based and adds noise to the input training data, making AE more robust for learning the input data.

### 3.2.3. POI Recommendation System Based on the Convolutional Neural Network (CNN) Method

Today, in pattern recognition methods and their apps, convolution neural network techniques are a great success in data analysis. Convolution neural network architecture mainly uses the relationship between some features or structural content and is at the center

of all techniques from Data Mining to predicting users visiting new POIs, recommender systems, and biological imaging [144–149].

This neuron-based network has a network-like topology that allows us to effectively extract key features and knowledge of POIs and friendships through passing via a series of kernel-sized convolution layers [150–152]. This model is based on neurons containing multiple trained biases and weights, and it is employed to extract, classify, and predict features. These biases and weights are used randomly at the beginning of the training. This method includes input layers, feature extraction (learning) layers, and classification layers.

The RecPOID model [3] proposes a novel deep-learning structure to achieve a true sequence of top-k of the best places to recommend to users. This novel model is a mix of convolution neural network (CNN) and fuzzy c-mean clustering technique that first specifies the closest friendship according to the behavioral pattern of user friend check-ins and clustering technique, and then the proposed convolution neural network model based on six features—user ID, month, day, hour, minute and second—predicts the subsequent location to visit according to the user's present location. Spatial analysis has been performed on clients' check-ins on well-known datasets Yelp [153] and Gowalla [154], encompassing many check-in data about geography. State-of-the-art algorithms UFC [155], LFBCA [49], and LORE [119] are investigated to validate the performance of the recommended RecPOID. The considerable accuracy of the proposed model was assessed using Precision and Recall. RecPOID consistently outperforms state-of-the-art algorithms. For instance, on Yelp, they obtain 0.037 for Precision@5 and 0.032 for Precision@10. This result indicates that the RecPOID technique was more effective than UFC, LFBCA, and LORE models of 0.01, 0.015, and 0.02, respectively. This performance improvement can be due to the integration of the clustering technique and friendship relations that have been effective in POI recommendations. The suggested RecPOID framework is shown in Figure 6. Zhai et al. [156] used the graph-based convolution neural network to build clients' check-in records on LBSNs to recommend the next potential POI for the target user.

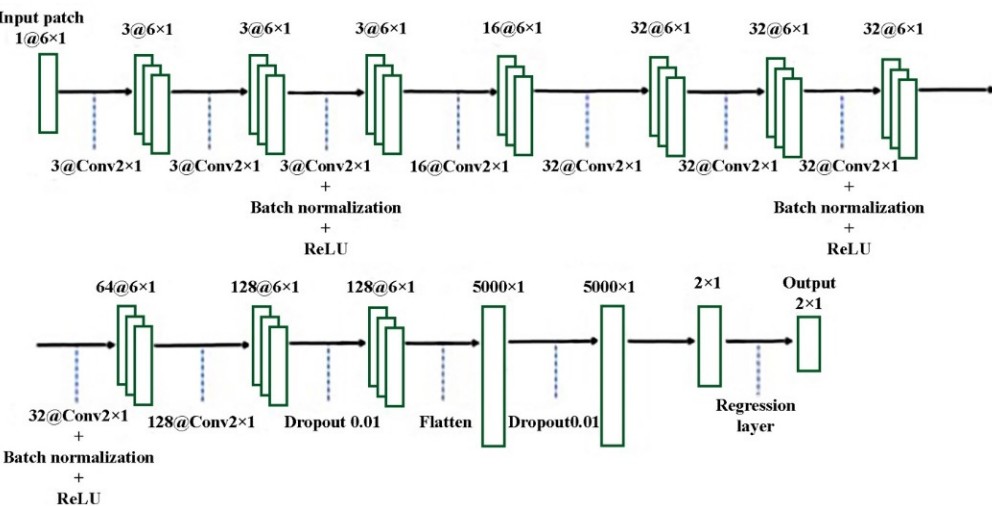

**Figure 6.** CNN model implemented in the RECPOID model [3].

3.2.4. POI Recommendation System Based on the Recurrent Neural Network (RNN) Method

RNN is a class of deep-learning architecture in which the features of each input data can have a weighting influence on the RNN output [136]. RNN runs on consecutive data that can be defined over a time series. For example, in speech recognition, entity recognition, translation, sentiment analysis, time-series issues, DNA sequencing, applications using video data, a number of natural language processing issues, recommender systems, etc. are applicable [157,158]. During the data flow for RNN, there must be coherence in the time axis. RNN operates in two forms: forward propagation and backpropagation. In recurrent neural network forward propagation, it is necessary to use a forward network to

share parameters with each other at any time step using the sequential structure of cells, and each output is generated under the influence of the previous parameters. In recurrent neural network backpropagation, the loss function must be calculated before performing the backpropagation calculations because in order to update the parameters, it is necessary to consider the derivatives of the loss function.

Zhong et al. [159] suggested a novel RNN-based deep neural network model called PDPNN to recommend POIs. This model learns dynamic user preferences by examining the user's long-term and short-term preferences. One part of the PDPNN considers the long-term preferences of the user from their check-in history and the other section considers short-term preferences with the help of his recent behavior. The similarity function introduced by this model examines the similarity between the spatio-temporal content of the user's current path and his past paths and suggests the appropriate POI according to the dynamic preferences.

The main field of application of recursive neural network algorithms is natural language processing using sequential word structure. RNNs suffer from the issue of data sparsity, which often exists in the field of investigation of recommender systems. Yang et al. [160] introduced a technique called Flashback for Sparse data by employing a recursive neural network method. This model attempts to decrease the disadvantages of the sparse datasets and offer more accurate suggestions. This article expressed that their approach could be applied to any RNN algorithm (Vanilla, LSTM, GRU, etc.). Essentially, providing spatial and temporal intervals to RNN as input is not an effective solution because users have certain behaviors, such as eating after work or exercising after leaving home on the weekends. This classical approach cannot literally learn this periodic behavior. Their flashback approach can arrange spatial and temporal interval factors for search with previously hidden states. Previous hidden states may have similar temporal–spatial patterns to current states. Using similar temporal–spatial patterns between the current and previous POIs, more accurate recommendations can be made for the next POI.

### 3.2.5. POI Recommendation System Based on a Long Short-Term Memory (LSTM) Method

LSTM [161] is a recurrent neural network method that has a single cell (memory) and can manage long sequence processes and transfer them to the next cell. Basic RNN cannot process long sequences. LSTM with three input, forget and output gates can set the cell state as a memory in the case of a long sequence and operate by transferring the current state to the next LSTM. LSTM can reduce the problem of vanishing gradients and exploding gradients of the RNN technique. In long sequence networks, there is an issue of vanishing gradient, and it is difficult to find the correct result (global minimum or maximum). In backpropagation of the training phase, the error will be differentiated and multiplied by each other; therefore, if the numbers are high, the exploding gradient problem arises.

Wang et al. [162] suggested a method for POI recommendation employing deep learning in LBSNs with respect to privacy. First, user information, relationships, and location information are reviewed. Then, based on the history of the user and the order of check-in's POIs, the LSTM mechanism is created, and the user information is used as input to obtain the short-term and long-term user preferences. Finally, social network knowledge and semantic knowledge are placed in various input layers, and to recommend the next POI to users, temporal and spatial information of user histories are used.

Sun et al. [163] introduced a long- and short-term preference modeling (LSTPM) approach for the next POI suggestion. In general, the dynamic behavior of users is presented under the two headings of long-term and short-term preference. Long-term movements are usually repetitive and generally do not show adaptability; on the other hand, the short term tends to be more variable. LSTPM, therefore, considers both long-term and short-term behaviors.

Doan et al. [164] suggested an Attentive Spatio-Temporal Neural model (ASTEN) in which LSTM is used along with an attention mechanism to recommend POIs. Thus, it is not merely dependent on the previous latent state to make offer users prefer to visit POIs

around them instead of places far away, and as the distance increases, the likelihood of visiting new POIs decreases.

### 3.2.6. POI Recommendation System Based on the Gated Recurrent Unit (GRU) Method

The gated recurrent unit (GRU) is a type of recurrent neural network [165]. This architecture is offered to address the weaknesses of the traditional RNNs such as the vanishing gradient problem as well as decreasing the overload in the LSTM architecture. GRU is generally considered to be a modified version of LSTM because both architectures use the same design. This model is simpler than LSTM but, in some cases, more useful. In this model, there are two gate operations, reset and update. Unlike LSTM, GRU has no output gate and outperforms better than LSTM in terms of speed, because it consumes less memory than LSTM. For long dependencies, it is recommended to use LSTM because of the extra memory so that the LSTM can be selected the first time and then switched to GRU.

RNN-based algorithms can be applied to long sequence structures and manage to retrieve information from these long structures. Short-term user preferences in recommendation problems are an issue that must be considered in developing an accurate recommendation system. The ATCA-GRU model [166] was proposed by Liu et al. to recommend the appropriate POI with respect to location classification and to reduce the impact of sparse user check-in data. To learn about the ATCA-GRU model, the authors incorporated categories related to the user's POIs and spatial–temporal information of his check-in data in the model. The Context-Category Specific Sequence Aware POI RS (CCS-POI-RS) was introduced using the Multi-GRU (MGRU) technique [167], which added two gates based on a specific context classification sequence that examines dynamic contexts and the influences of transition contexts. The contextual attention recurrent architecture (CARA) model [168] introduced two additional gating mechanisms (contextual attention gate (CAG) and time- and spatial-based gate (TSG)). They explore the status of the previously hidden state based on time and geographical differences between the two POIs.

### 3.2.7. POI Recommendation System Based on Deep Reinforcement Learning Method

Deep reinforcement learning (RL) [169] is a subset of machine learning that incorporates reinforcement learning and deep learning. This is a popular model used today in fields such as games [170], autorun machines and robotics [171], and networking [172], and it is used in recommendation systems to rank target POI scores for the intended user, and sequential recommendations [173]. This technique has powerful representational learning and performance approximation features to meet the challenges of artificial intelligence [174], and even without manual state space engineering, it can help software to make decisions with unstructured input data and learn how to reach their goals. The entire framework consists mostly of the following components: environments, agents, states, rewards, and actions. This means that the function's approximation and the goal optimization unite the mapping of states and actions into the rewards that lead to it. The purpose of reinforcement learning is to find a recommendation strategy which can be best rewarded.

The DeepPage method [175] modeled a DRL-based recommendation system to provide solutions for issues. This technique can effectively update the recommendation strategy in real time. The authors took the page recommendations for this research and developed a model that can optimize a page of items with the proper display based on real-time user feedback. This method can be used for future work in the field of POI recommendations.

The Table 1 summarizes the new approaches introduced by recent studies on POI recommendation and deep-learning technology. In fact, these models are a combination of traditional machine-learning techniques and deep learning. The datasets used in these selected studies as well as solution evaluation metrics are also determined.

**Table 1.** Review of methods, datasets, and evaluation metrics of recent studies in the field of POI recommendation systems based on deep-learning methods.

| Recent Models | | Methods | | | | | | Datasets | | | | | | | | | Validation Metrics | | | | | | | | |
|---|---|---|---|---|---|---|---|---|---|---|---|---|---|---|---|---|---|---|---|---|---|---|---|---|---|
| Year | System | MLP | AE | CNN | RNN | LSTM | GRU | Foursquare | Gowalla | Yelp | Instagram | Tencent | Brightkite | Twitter [176] | Yahoo! Flickr | Yahoo! Japan | Prec | Rec | F1 | RMSE | MAP | MRR | NDCG | AUC | Logloss |
| 2021 | POIRA [69] | | ✓ | ✓ | | | | | | | ✓ | | | | | | ✓ | ✓ | ✓ | | | | | | |
| 2021 | POI-LSA [162] | | | ✓ | | ✓ | | | | ✓ | | | ✓ | | | | ✓ | ✓ | | | | ✓ | | | |
| 2021 | RecPOID [3] | | | ✓ | | | | | ✓ | ✓ | | | | | | | ✓ | ✓ | | | | ✓ | | | |
| 2021 | Deep-RegionRS [177] | ✓ | | ✓ | | ✓ | ✓ | | | | | | | | | | | | | | ✓ | | | | |
| 2021 | PBGCN [156] | | | ✓ | | | | | | | | | ✓ | | | | ✓ | | | | | | | | |
| 2021 | ATCA-GRU [166] | | | | | | ✓ | ✓ | | | | | | | | | ✓ | ✓ | ✓ | | | | | | |
| 2021 | Bi-LSTM-Attention [178] | | | | | ✓ | | ✓ | ✓ | | | | | | | | ✓ | ✓ | ✓ | | | | | | |
| 2021 | ST-PIL [179] | ✓ | | | | ✓ | ✓ | | | | | | | | | | ✓ | | | | | | ✓ | | |
| 2021 | LSVP [180] | ✓ | | ✓ | | ✓ | | | | | | | | | ✓ | | ✓ | | | | | | | ✓ | |
| 2021 | HOPE [138] | ✓ | | | | ✓ | ✓ | | | ✓ | | | | | | | ✓ | | | | | ✓ | | | |
| 2021 | ConvLSTM [177] | ✓ | | | | ✓ | ✓ | | | | | | | | | | | | | | ✓ | | | | |

**Table 1.** *Cont.*

| Recent Models | | Methods | | | | | | Datasets | | | | | | | | | Validation Metrics | | | | | | | | |
|---|---|---|---|---|---|---|---|---|---|---|---|---|---|---|---|---|---|---|---|---|---|---|---|---|---|
| Year | System | MLP | AE | CNN | RNN | LSTM | GRU | Foursquare | Gowalla | Yelp | Instagram | Tencent | Brightkite | Twitter [176] | Yahoo! Flickr | Yahoo! Japan | Prec | Rec | F1 | RMSE | MAP | MRR | NDCG | AUC | Logloss |
| 2021 | DSPR [22] | | | | | ✓ | | ✓ | ✓ | | | ✓ | | | | | | ✓ | | | ✓ | | | | |
| 2020 | DeepSTIN [181] | | | | | ✓ | | | | | | | | | | ✓ | | | | | | | | ✓ | ✓ |
| 2020 | PDPNN [159] | | | | | ✓ | | ✓ | | | | | | | | | | ✓ | | | | | | | |
| 2020 | Flashback [160] | | | | ✓ | | | ✓ | ✓ | | | | | | | | ✓ | | | | | | ✓ | | |
| 2020 | LSTPM [163] | | | | | ✓ | | ✓ | ✓ | | | | | | | | | ✓ | | | | | | ✓ | |
| 2020 | PGIM [182] | ✓ | | ✓ | | | | ✓ | ✓ | ✓ | | | | | | | ✓ | ✓ | | | | | | ✓ | |
| 2020 | HGMAP [183] | ✓ | | ✓ | | | | ✓ | ✓ | ✓ | | | | | | | ✓ | ✓ | | | | ✓ | | | |
| 2020 | GLR-GT-LSTM [184] | | | | | ✓ | | ✓ | ✓ | | | | | | | | ✓ | ✓ | | | | | | | |
| 2019 | STGCN [185] | | | | | ✓ | | ✓ | ✓ | | | | ✓ | | | | ✓ | | | | | ✓ | | | |
| 2019 | Real-time MF [186] | | | ✓ | | | | ✓ | | | | | | | ✓ | | | ✓ | | | | | ✓ | | |
| 2019 | ASTEN [164] | | | | | ✓ | | ✓ | ✓ | | | | | | | | | ✓ | ✓ | | | | | ✓ | |
| 2019 | MGRU [167] | | | | | | ✓ | ✓ | ✓ | | | | | | | | | ✓ | | | | | ✓ | | |

**Table 1.** *Cont.*

| Recent Models | | Methods | | | | | | Datasets | | | | | | | | | Validation Metrics | | | | | | | | |
|---|---|---|---|---|---|---|---|---|---|---|---|---|---|---|---|---|---|---|---|---|---|---|---|---|---|
| Year | System | MLP | AE | CNN | RNN | LSTM | GRU | Foursquare | Gowalla | Yelp | Instagram | Tencent | Brightkite | Twitter [176] | Yahoo! Flickr | Yahoo! Japan | Prec | Rec | F1 | RMSE | MAP | MRR | NDCG | AUC | Logloss |
| 2019 | CPC [187] | | | √ | | | | √ | | | | | | | | | √ | √ | | | | | | | |
| 2018 | PEU-RNN [188] | | | | √ | | | | √ | | | | √ | | | | √ | √ | | | | | | | |
| 2018 | SAE-NAD [142] | | √ | | | | | √ | √ | √ | | | | | | | √ | √ | | | √ | | | | |
| 2018 | ReEl [189] | | | √ | | | | | | √ | | | | | | | √ | √ | √ | | | | | | |
| 2018 | CARA [168] | | | | √ | | √ | √ | | √ | | | √ | | | | | | | | | | | √ | |

### 3.3. Advantages and Disadvantages of Machine Learning and Deep Learning Models

Machine-learning and deep-learning techniques have the benefits and weaknesses described below, and according to these characteristics, researchers can achieve good performance in various issues.

In the collaborative filtering model, if there is not sufficient data on the user or item, this technique suffers from the problem of cold start; and if the database is sparse, this model does not have the ability to make proper recommendations. This method has issues in terms of memory consumption and time complexity because it performs on a large matrix of user–item interaction. No additional plugins are possible for model-based collaborative filtering. Therefore, this approach alone is no longer sufficient for complex issues. In POI recommendation systems, the collaborative filtering method has recently been used to solve part of the problem. For example, researchers can use the collaborative filtering method to preprocess their datasets and use this technique as part of a solution based on the deep-learning model. Today, this model alone is not enough to recommend a POI.

Another recommendation technique in POI recommending systems is the Markov chain, which is often used to build the probabilities of different states and the transition rates between them. The Markov chain model can be easily implemented. This method is very useful for modeling the stochastic process of discrete time and discrete state space in different fields such as finance (stock price movement), NLP algorithms (finite state converters, hidden Markov method for POS tagging), or even in engineering physics. Markov models can also be used for pattern recognition, predicting, and learning sequential data statistics, but Markov chain models are unsuitable for examining short time intervals because individual displacements are not random but are related definitely in time. This method is not used to collect temporary user preferences; for this reason, it cannot recommend personal POIs. Predicting the client's next POI depends only on the previous check-in event. This ignores the client's long-term and short-term preferences.

Matrix factorization and its derivatives are commonly used as a representational approach to user modeling and are suitable for finding hidden relationships between the user and the item, but alone they do not solve complex problems such as POI recommendation. Traditionally, it is two-dimensional and unsuitable for multidimensional data; therefore, the Tensor Factorization approach [28] has been developed to solve this problem. Probabilistic Matrix Factorization [190] is also used to correlate the user and the item and to address the cold start challenge. Matrix factorization has sparsity, cold start, and scalability problems. Of course, this method can free up a significant amount of memory by reducing the dimension.

A multilayer perceptron (MLP) contains one or several hidden layers (other than input and output layers) and can learn linear and nonlinear functions. This approach is suitable for prediction issues and is used to provide predictions of new POIs. Based on the data provided for learning—the ability to learn how to perform tasks or initial experience—MLPs are capable of generalizing; this means that they classify an unrecognized pattern with other recognized patterns that have the same distinctive features, such as noisy or insufficient inputs. However, this method contains many parameters because it has a fully connected structure and leads to redundancy and inefficiency. Most articles have used combined methods to solve POI recommendation problems and have applied this approach in their prediction phase.

The autoencoder approach enhances performance and provides a data-based model instead of predefined filters. Autoencoders offer filters that may generally better match the problem data. This method technically works better on dirty data but may delete important information in the input data. The autoencoder algorithm requires a target function to evaluate the accuracy of the encoder/decoder input data.

The convolutional neural network (CNN) has easy understanding, fast execution, and the highest accuracy among all algorithms that predict images. This network can acquire local and global features, thus greatly improving the efficiency and accuracy of the model.

It is very powerful in processing unstructured multimedia information. Convolutional neural networks can be used to extract image features and examine the effect of visual features on POI recommendation systems. CNN can be used to extract features from the text. Graph-based CNNs can perform interactions on recommended tasks. These networks provide good spatial connections. Automatic detection of important features without human supervision is a prominent characteristic of these networks, but CNN does not encode the position and orientation of objects and instead requires a lot of training data. If CNN has multiple layers, it needs a computer with a good GPU; otherwise, the training process will take a long time.

In many applications, the issue of time is a determining factor, so the output of the system at any time is a function of its input and also the output of the system in previous times. In some cases, system output may even depend on system input at earlier times. To model such systems by neural networks, time representation in the operation of these networks is inevitable. Time can be implemented in neural networks in two ways: direct and implicit. One way to implicitly represent time is to use RNN. Past information (times) is considered in computations, and weights are shared over time and are useful in predicting time series. Another advantage is that input processing is able with any length. Model size does not increase with input size, but the computation in this model is slow, and old information is difficult to access. Moreover, we do not consider subsequent inputs in the current state. Another limitation is the vanishing gradients problem, which complicates the training activity.

The long short-term memory (LSTM) network is one of the most important solutions for overcoming the vanishing gradient problem. LSTM performs this through forget, input, and output gates. The forget gate ignores the amount of available memory, the input gate determines what appropriate information can be added from the existing step, and the output gate specifies the value of the following hidden state. This network has the ability to maintain memory/state from previous activations rather than total activations and can remember features for a long time. LSTM operates more accurately in datasets with a longer sequence. LSTMs provide a wide range of parameters such as learning rate and input and output biases. Thereupon, there is no need for accurate adjustments. However, this network has constraints for modeling continuous spaces and consumes high memory.

Gated recurrent unit (GRU) models, such as LSTM, can maintain memory/state from previous activations instead of total activations compared to RNN and can be used to solve the vanishing gradient problem. The GRU network, compared to other models including LSTM, exposes the cell state to other network units and performs both input and output operations via its reset gate. GRU uses fewer training parameters, so it consumes less memory and runs faster.

Reinforcement machine-learning algorithms make it possible to model various additional information to design the proposed strategies in real time, and due to the real-time feedback and production of a page of items with appropriate representation, they can be used in POI recommendation systems. This technique shows how useful or harmful a particular action is in relation to a particular situation, and gains its knowledge based on its success or failure. However, this algorithm is not suitable for solving simple problems and requires a lot of data and computations. Excessive reinforcement learning can lead to the overload of states, which can reduce results.

## 4. The Most Widely Used Evaluation Metrics and Datasets

In this section, the metrics and datasets that are most frequently utilized today to evaluate POI recommendations based on deep learning are discussed and the third question of the article is answered.

### 4.1. Review of Evaluation Metrics

The purpose of this section is to review the evaluation metrics that most authors of recent state of the art have used to measure their POI recommender systems based on deep learning. These metrics have been observed at least once in selected studies. Most articles use accuracy, precision, recall, and other error-based metrics that are based on effectiveness, and this indicates that recent investigations have prioritized accuracy. After them, the MAP evaluation metric has more utilization in approaches and is used to evaluate object detection methods. According to Table 1, most LSTM-based studies have used this metric. F1 and NDCG are also very significant. F1 is suitable for evaluating binary classification systems and categorizes samples as positive and negative. NDCG metric is employed in recommender systems in various fields and determines how well the system is working. It is a popular way to measure the quality of a set of search outcomes and match the rating submitted by the user with the ideal rating. These iterative metrics can help to compare and replicate the results and can evaluate different research solutions. The Table 2 provides a brief description of the most widely used evaluation metrics.

**Table 2.** A short description of the most widely employed evaluation metrics in selected research.

| Evaluation Metrics | Brief Introduction |
|---|---|
| Prec (Precision) | A fraction of the recommended top-k POIs have been successfully recommended to the target customer and it is defined as the ratio of recommended POIs to the number of related recommended POIs. |
| Rec (Recall) | A fraction of K is a well-recommended POI visited by the intended user and a ratio of recommended and related POIs to the number of related POIs is defined. |
| F1 (F-measure) | It is a precision and recall combination and is obtained by calculating the weighted harmonic mean between these two criteria. |
| RMSE (Root Mean Squared Error) | The root-mean-square error calculates the distinction between the actual and estimated POIs (predicted) by the recommender system. |
| MAP (Mean Average Precision) | Calculate the mean values of the average precision according to all the recommendations lists created for the customers. |
| MRR (Mean Reciprocal Rank) | The criteria for evaluating systems is that returning a ranked list of answers to queries, and the inverse multiplication of the rank of the first answer is correct. |
| NDCG (Normalized Discounted Cumulative Gain) | It is a normalized DCG technique. The DCG is an accuracy measure based on the position of the ranking position and examines a list of recommendations according to the relevance of the ranking position. |
| AUC | It is used for classification topics and the area under the curve can be calculated by Simpson's law. This criterion estimates the probability that a classifier will select a randomly selected positive sample above a random negative one. |
| Log loss | It is appropriate when the output of the model is likely to be a binary result. Evaluates performance by comparing actual location tags and predicted probabilities. This comparison is measured using cross-entropy and quantifies classification accuracy by penalizing incorrect classifications. |

### 4.2. Review of Datasets

Most of the datasets investigated in the selected studies are Foursquare, Gowalla, and Yelp, which are very popular and practical in POI recommendations. A short description of these datasets is in the Table 3.

**Table 3.** A brief description of the most popular datasets employed in selected research.

| Datasets | Brief Introduction |
| --- | --- |
| Foursquare [191] | It is a location-based service where customers can share their POIs with friends and collect prize, badges, and coupons. This dataset includes check-ins related to Tokyo and New York cities. |
| Yelp [153] | It is a location-based social network (LBSN) that places businesses such as hotels, shopping malls, and restaurants. Users at Yelp check in with various vendors to comment and rate these POIs. |
| Gowalla [154] | This dataset is an LBSN where users share their POI information with friends by check-in. |
| Instagram [69,192] | This is a social network that allows any user to link geo-tags to their images and comments. |
| Brightkite [193] | It is a location-based social networking service where customers share their check-in POIs with friends and users' social information is available. |
| Tencent [194] | It is a China-based mobile check-in service that provides social network information and spatial-temporal history to users. It can be used to analyze users' personal and social information and check-in records along with their time. |
| Yahoo! Flickr [195,196] | This database is the largest public multimedia collection that includes uploaded images and videos with related information including geographical coordinates, history, and geographical accuracy. |
| Yahoo! Japan [197] | This application is related to Yahoo, which provides online shopping services for map search engines, shipping applications, and other services. With the location service of this software, the POI status of users can be checked according to their activity time. |

We suggest a new pipeline named DeePOF [148] regarding POI recommendations and deep learning. The purpose of this technique is to gain the proper top-K point-of-interest sequence per customer. Our method utilizes the novel convolutional neural network and mean-shift clustering technique. We offer effective recommendations to users based on geographical and temporal information, as well as behavioral information from close friends. This hybrid technique is evaluated on two real datasets, Yelp and Gowalla of LBSNs. Six state-of-the-art approaches, UFC [155], LFBCA [49], LORE [119], HGMAP [183], APOIR [198], and SAE-NAD [142], are selected for DeePOF performance validation and have been validated with two criteria Recall @ K and Precision @ K. The suggested prediction method results on Yelp and Gowalla datasets are represented in Figure 7.

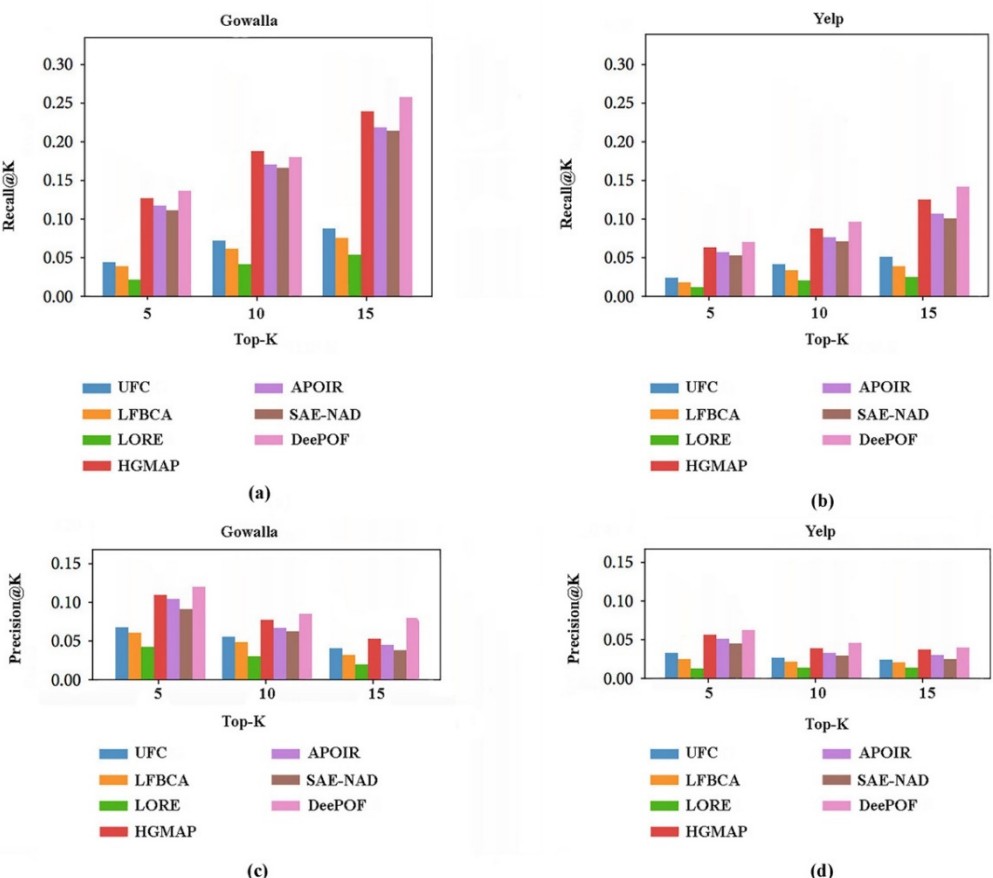

**Figure 7.** DeePOF [148] compared to 6 other models on the Gowalla and Yelp datasets. (**a**) Recall on the Gowalla. (**b**) Recall on the Yelp. (**c**) Precision on the Gowalla. (**d**) Precision on the Yelp.

## 5. Conclusions

This is a systematic review aimed at reviewing common deep-learning methods in POI recommendation systems and discussing the most prominent recent techniques in this field. In this research, several questions are presented, and they are answered in different sections of the article. Recommender systems and influential factors and the challenges were expressed. Machine-learning and deep-learning models were introduced in this area, and their advantages and disadvantages were compared. The datasets and evaluation metrics used in recent approaches were categorized, and the most important future directions and open problems were presented. The principal findings of this research can be outlined as follows:

- Many recent approaches on POI recommendations have used combination methods with deep-learning models to increase the accuracy of their recommendations and try to improve their suggestions to users;
- Geographic data, the importance of similar friendships, user preferences, and temporal data are the most commonly used factors in modeling solutions related to POI recommendations;
- The most widely used deep-learning methods in POI recommendation are LSTM and CNN, and the use of collaborative filtering models, matrix factorization, and its derivatives are popular along with another model;
- Most of the datasets used in recent studies in the field of deep-learning-based POI recommendation have been extracted from data related to well-known LBSNs such as Foursquare, Gowalla and Yelp Moreover, evaluation metrics for comparison between approaches are mostly based on precision and recall measures, which illustrates the attention to accuracy.

The most significant future directions and open issues in response to question 4 are summarized as follows:

- To make progress towards more useful POI recommendation systems and to promote high quality research, authors should present their problem-oriented strategy with a combination of up-to-date methods according to the audience and their concerns and needs and offer a proper solution.
- Researchers can develop a hybrid structure that incorporates the advantages of each technique and limits their disadvantages.
- Developing models to improve the realization of user behavior by incorporating influential factors such as geographical, temporal, social factors, textual data, sentiment analysis of images and videos, upcoming events, weather, traffic, use of IoT components, and sensors can be effective, and various methods such as natural language processing (NLP) and topic modeling help to create novel POI recommendation techniques. Some of these extra features may be useful to the user, some may be more appealing to POI owners, and depending on the type of issue, the best factors can be used.
- Users may have different choices depending on their situation in the city or or their traveling situation. Restrictions on the user's choice of POI can be significant—factors such as a disease outbreak, telecommuting, the maximum travel length, the price of attractions, the unexpected crisis, staff strikes, and political conditions. It is recommended to use a more diverse dataset that examines the user from different angles because the user may have checked in under the conditions created by the system.
- In addition to accuracy, other dimensions of quality such as novelty and diversity can be considered in evaluations.
- Recommending system security and users' privacy, paying attention to ethical issues, and preventing the misuse of users' information are also important problems that should be investigated, and providing personal advice and hiding users' sensitive information can be influential.
- Another issue is that few studies have demonstrated the code and how to preprocess data and implement them, so publishing this content realistically could help us to analyze the models and solutions better.

This systematic review is hoped to be useful for researchers and businesses interested in studying POI recommendations.

**Author Contributions:** Conceptualization, S.S., M.J. and M.H.; methodology, S.S., M.J. and M.H.; formal analysis, S.S., M.J. and M.H.; data curation, S.S., M.J. and M.H.; writing—original draft preparation, S.S., M.J. and M.H.; writing— review and editing, S.S., M.J. and M.H. All authors have read and agreed to the published version of the manuscript.

**Funding:** This research received no external funding.

**Informed Consent Statement:** Informed consent was obtained from all subjects involved in the study.

**Data Availability Statement:** Data is contained within the article.

**Acknowledgments:** Authors acknowledge Support by the KIT-Publication Found of the Karlsruhe Institute of Technology, Germany.

**Conflicts of Interest:** The authors declare no conflict of interest.

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
