# Peer review of "Toward Point-of-Interest Recommendation Systems: A Critical Review on Deep-Learning Approaches"

_electronics, doi:10.3390/electronics11131998_

Round 1

Reviewer 1 Report

This paper seems to be fair. I recommend to accept the manuscript with the following minor corrections.

In the Overview of POI Recommender Systems, the authors have reviewed the previous works.

However, the issues and drawbacks of the previous works are not point out properly. Previous works pros and cons should be discussed well and pointed out along with various filtering techniques.

Problem statement should be included under a separate sub-headings.

The accuracy of CNN model implemented in the RECPOID model should be discussed along with comparison of other models used till date. 

Conclusion section the principal findings should be revised. 

Author Response

Toward Point-of-Interest Recommendation Systems: a Critical Review on Deep Learning Approaches

Journal of electronics

Response to the Reviewers’ Comments

The reviewers’ high quality of refereeing helped us to clarify many areas in the paper. We greatly appreciate the editor-in-chief and reviewers again for giving us an opportunity to revise our manuscript. We incorporated the revisions recommended by the reviewers and carefully considered the reviewer comments which are attached. We provide below our responses and detailed revisions we have made to every comment that the reviewer has provided. We highlighted all revisions in the revised manuscript (Highlighted in yellow). Detailed descriptions are given in this revision note below.

Point-to-Point responses to the Reviewer.

Reviewer #1:

This paper seems to be fair. I recommend to accept the manuscript with the following minor corrections.

  1. In the Overview of POI Recommender Systems, the authors have reviewed the previous works.

However, the issues and drawbacks of the previous works are not point out properly. Previous works pros and cons should be discussed well and pointed out along with various filtering techniques.

RESPONSE:

We are grateful to you for your insightful and helpful comments. In the overview of POI recommending systems, the previous works were discussed as follows:

CFSKW model [40] proposes a POI recommendation technique that utilizes geographical influence and the role of geography in the recommendation is mentioned separately.  The quality of the recommendation is improved by integrating collaborative filtering and a method for spatial kernel weighting. The two important factors that are tuned to the CFSKW model are the kernel bandwidth and the coefficient of incorporating user preference with geographical impact. The suggested method experimented on New York and Tokyo Foursquare datasets. According to the findings of this study, low bandwidth as well as medium to high geographical coefficient yield more accurate results. A proposed dynamic bandwidth-based algorithm has shown promising outcomes at disproportionate densities of POIs, such as the Tokyo dataset. The proposed method has been evaluated with state-of-the-art approaches and analyzed with Precision@K and  Recall@K metrics. This evaluation indicates the CFSKW method in terms of Precision@5 and Recall@5 respectively, 3% and 5.1% in the New York dataset also 1.1% and 1.2% in the Tokyo dataset performed better than the reference algorithms, and the use of the dynamic bandwidth geographic similarity technique has been effective in improving performance.

----------------

The paper [60] suggested an algorithm for online marketing recommendations that integrate content and collaborative filtering. This fusion recommendation technique solves the new client problems by relying upon content filtering, sparsity of data based on collaborative filtering, and cold start challenge problems. The interests of existing users are first discovered. Then the potential interest of the client is extracted from the model of combined similarity of content and behavior by considering the similar user group of the target user and predicting the interest of the user for feature words. After that, the existing and potential interests of the user are combined. Finally, to provide appropriate recommendations, the similarity between the content and the fusion technique is estimated and then clustered via K-means. This article has been evaluated for the MovieLens data set by Recall, Precision, and hybrid similarity metrics. The proposed hybrid method can solve the mentioned challenges and has a great effect in terms of recall, accuracy, and diversity.

----------------

In this paper [47], a novel recommendation algorithm in the field of IoT and remote device control with the help of smartphones is proposed, which is a combination of content-based and collaborative filtering methods and uses context information such as orientation and location. Other tasks considered in this work are location detection, ambiguity handling, recommendation making, orientation detection, item extraction, and profile creation. This proposed method performs better in terms of the level of personalization due to the focus on the role of user orientation. Future studies can be applied to location-based recommendations such as restaurants and rest spots. Increasing user privacy is also a very important issue.

  1. Problem statement should be included under a separate sub-headings.

RESPONSE:

We are thankful to the reviewer for the thorough and constructive comments. Based on your guidance, we placed the Problem statement under a separate sub-heading.

3.The accuracy of CNN model implemented in the RecPOID model should be discussed along with comparison of other models used till date. 

RESPONSE:

Based on your valued and effective points, the following content is added to the RecPOID description:

Spatial analysis has been performed on users’ check-ins on public datasets Yelp and Gowalla, encompassing many check-in information about geography. State-of-the-art algorithms UFC, LFBCA, and LORE are investigated to validate the performance of the recommended RecPOID. The considerable accuracy of the proposed model was assessed using Precision and Recall. RecPOID consistently outperforms state-of-the-art algorithms. For instance, on Yelp, they obtain 0.037 and 0.032 in terms of Precision@5 and Precision@10, respectively. It means RecPOID method exhibiting 0.01, 0.015, and 0.02 performance improvement compared with UFC, LFBCA, and LORE, respectively. This superiority is perhaps outlined by the fact that the suggested RecPOID completely utilized the combination of the clustering method and friendship relations to recommend point-of-interests. The proposed RecPOID framework is illustrated in Figure 4.

  1. Conclusion section the principal findings should be revised. 

RESPONSE:

We are thankful to the reviewer for the thorough and constructive comments. Based on your guidance, the conclusion is improved as follows:

This is a systematic review aimed at reviewing common deep learning methods in POI recommendation systems and discussing the most prominent recent approaches in this field. In this research, several questions are presented and they are answered in different sections of the article. Recommender systems and influential factors and the challenges were expressed. Machine learning and deep learning models were introduced in this area and compare their advantages and disadvantages. The datasets and evaluation metrics used in recent approaches were categorized and the most important future directions and open problems were presented. The principal findings of this research can be outlined as follows:

  • Many recent approaches on POI recommendations have used combination methods with deep learning techniques to increase the accuracy of their recommendations and try to improve their suggestions to users;
  • Geographic data, the importance of similar friendships, user preferences, and temporal data are the most commonly used factors in modeling solutions related to POI recommendations;
  • The most widely used deep learning methods in POI recommendation are LSTM and CNN and the use of collaborative filtering techniques, matrix factorization, and its derivatives are popular along with other model;
  • Most of the datasets used in recent studies in the field of deep learning-based POI recommendation have been extracted from data related to well-known LBSNs such as Foursquare, Gowalla and Yelp Also, evaluation metrics for comparison between approaches are mostly based on Precision and Recall measures, which illustrates the attention to accuracy.

The most significant future directions and open issues in response to question 4 are summarized as follows:

  • To make progress towards more useful POI recommendation systems and to promote high quality research, authors should present their problem-oriented strategy with a combination of up-to-date methods according to the audience and their concerns and needs and offer a proper solution.
  • Researchers can develop a hybrid structure that incorporates the advantages of each technique and limits their disadvantages.
  • Developing models to improve the realization of user behavior by incorporating influential factors such as geographical, temporal, social factors, textual data, sentiment analysis of images and videos, upcoming events, weather, traffic, use of IoT components, and sensors can be effective and various methods like natural language processing (NLP) and topic modeling help to create novel POI recommendation techniques. Some of these extra features may be useful to the user, some may be more appealing to POI owners that depending on the type of issue, the best factors can be used.
  • Users may have different choices depending on their situation in the city or traveling. Restrictions on the user's choice of POI can be significant; Factors like a disease outbreak, telecommuting, the maximum travel length, the price of attractions, the unexpected crisis, staff strikes, and political conditions. It is recommended to use a more diverse dataset that examines the user from different angles, because the user may have checked-in under the conditions created by the system.
  • In addition to accuracy, other dimensions of quality like novelty and diversity can be considered in evaluations.
  • Recommending system security and users' privacy, paying attention to ethical issues, and preventing the misuse of users' information are also important problems that should be investigated, and providing personal advice and hiding users' sensitive information can be influential.
  • Another issue is that few studies have demonstrated the code and how to preprocess data and implement them, so publishing this content realistically could analyze the models and solutions better.

Once again, we would like to express our great appreciation to the editor and the reviewer for their careful readings and valuable comments on our paper. Looking forward to hearing from you.

Sincerely,

Sadaf Safavi, Mehrdad Jalali, Mahboobeh Houshmand

Reviewer 2 Report

Title: Toward Point-of-Interest Recommendation Systems: A critical review on deep learning approaches

Comments:

With the development of Internet infrastructure and deep learning, Location-Based Social networks (LBSNs) and various deep learning methods have been proposed, which create a variety of new point-of-interested (POI) recommender system algorithms. This paper focuses on reviewing POI recommender systems in the decade. But I still have the following concerns.

1. Section 1: The corresponding abbreviations should be defined when they are used first and later they can instead of the full spelling. Besides, the research questions at the end are insufficient to generalize the contribution of this article. Please conclude the contribution of this paper.

2. Section 2: there are sub-title errors in this subsection. Furthermore, basic recommendation algorithms (e.g., collaborative filtering, etc.) and specific application scenarios (e.g., social recommender systems, etc.) should be described under different sub-title to show the difference. It would be better to introduce the influence factors of POI recommendations before introducing the challenges of POI recommender system.

3. Section 3: spelling or proper noun errors such as the Learning Machine on line 349 need to be checked. It is better to analyze specific algorithms and explain the advantages and disadvantages of the different methods.

4. Section 4: this section just briefly describes some evaluation metrics and datasets. First, It is better to conclude the application scenarios of the different evaluation metrics. Second, it is better to provide the download link of the datasets.

5. Section 5: the content of future work directions is not clear enough. It is better if the future research directions are listed for the soundness of readability. Besides, please describe the method of improving the model and future application scenarios in different paragraphs.

6. It is better to compare some representative models. Such comparisons could be done by citing some state-of-the-art records reported by recent publications or conducting some experiments by the authors. 7. Literature review. Some related studies are missing. For example, latent factor analysis is a critical branch of recommendation. It is efficient in addressing the sparse issue of recommendation. It is better to discuss latent factor analysis-based recommender systems. There are many related papers, such as 1) A Data-Characteristic-Aware Latent Factor Model for Web Service QoS Prediction, IEEE Transactions on Knowledge and Data Engineering, vol. 34, no. 6, pp. 2525-2538, 2022. 2) An L1-and-L2-norm-oriented Latent Factor Model for Recommender Systems, IEEE Transactions on Neural Networks and Learning Systems, DOI: 10.1109/TNNLS.2021.3071392. 3) A posterior-neighborhood-regularized latent factor model for highly accurate web service QoS prediction, IEEE Transactions on Services Computing, vol. 15, no. 2, pp. 793-805, 1 March-April 2022. 4) A Deep Latent Factor Model for High-Dimensional and Sparse Matrices in Recommender Systems, IEEE Transactions on Systems Man and Cybernetics: Systems, vol. 51, no. 7, pp. 4285-4296, 2021, DOI:10.1109/TSMC.2019.2931393. 5) Di Wu, Peng Zhang, Yi He, and Xin Luo, A Double-Space and Double-Norm Ensembled Latent Factor Model for Highly Accurate Web Service QoS Prediction, IEEE Transactions on Services Computing, 2022, doi: 10.1109/TSC.2022.3178543.

8. Language issues, typos, and grammar errors.

Author Response

Toward Point-of-Interest Recommendation Systems: a Critical Review on Deep Learning Approaches

Journal of electronics

Response to the Reviewers’ Comments

The reviewers’ high quality of refereeing helped us to clarify many areas in the paper. We greatly appreciate the editor-in-chief and reviewers again for giving us an opportunity to revise our manuscript. We incorporated the revisions recommended by the reviewers and carefully considered the reviewer comments which are attached. We provide below our responses and detailed revisions we have made to every comment that the reviewer has provided. We highlighted all revisions in the revised manuscript (Highlighted in yellow). Detailed descriptions are given in this revision note below.

Point-to-Point responses to the Reviewer.

Reviewer #2:

With the development of Internet infrastructure and deep learning, Location-Based Social networks (LBSNs) and various deep learning methods have been proposed, which create a variety of new point-of-interested (POI) recommender system algorithms. This paper focuses on reviewing POI recommender systems in the decade. But I still have the following concerns.

  1. Section 1: The corresponding abbreviations should be defined when they are used first and later they can instead of the full spelling. Besides, the research questions at the end are insufficient to generalize the contribution of this article. Please conclude the contribution of this paper.

RESPONSE:

We are grateful to you for your insightful and helpful comments. Abbreviations reviewed and corrected. Also, according to your guidance, the research questions in the following were corrected and the contribution of this article was added:

In this work, our research questions are the following:

RQ1. What are the types of POI Recommender Systems, influencing factors, and challenges in their recommendation?

RQ2. What are the traditional machine learning methods and deep learning models that have been considered in recent POI recommender systems?

RQ3. What are the most widely employed evaluation metrics and popular datasets to evaluate POI recommendations based on deep learning?

RQ4. what are the most significant future research trends and open issues?

Contributions of this survey

The goal of this systematic literature review is to specify the recent state-of-the-art in the field of deep learning-based POI recommendations. This work can contribute to the success of research areas in universities and industrial centers and researchers with rich knowledge of the factors influencing POI recommender systems and traditional machine learning approaches, can select appropriate deep neural networks and combine different methods for solving their proposed tasks. Authors can consider the strengths and weaknesses of methods, review evaluation criteria, and popular datasets. This paper provides an overview of the state-of-the-art and identifies the new trends and future directions in this research field.

  1. Section 2: there are sub-title errors in this subsection. Furthermore, basic recommendation algorithms (e.g., collaborative filtering, etc.) and specific application scenarios (e.g., social recommender systems, etc.) should be described under different sub-title to show the difference. It would be better to introduce the influence factors of POI recommendations before introducing the challenges of POI recommender system.

RESPONSE:

We are thankful to the reviewer for the thorough and constructive comments. Based on your guidance, we corrected the sub-titles. The influence factors of the POI recommendation were placed before introducing the challenges of the POI recommender system.

  1. Section 3: spelling or proper noun errors such as the Learning Machine on line 349 need to be checked. It is better to analyze specific algorithms and explain the advantages and disadvantages of the different methods.

RESPONSE:

Thank you for pointing to these items. spelling or proper noun errors were checked and corrected. Further details about a number of algorithms were also added to the article, such as the following:

PR-RCUC [26] model proposes a novel POI recommendation technique that integrates the region-based collaborative filtering method with a user-based mobile context. A challenge in this work is data sparsity and also it is difficult to provide a logical explanation to the user in relation to the suggestion and visit to the desired location. To reduce the data sparsity problem, this model first clusters spots in various regions and combines the region factor with the collaborative filtering method. The next task of this technique is to create the mobile context for a client like geographical distance and categories of the location. Finally, it fuses the above two parts and presents the PR-RCUC method. In this model, two datasets from Foursquare are examined and also employ three widely used criteria of Accuracy, Recall and F1-score to evaluate the proposed model. Experimental results indicate the PR-RCUC algorithm outperforms some famous recommendation algorithms. In the continuation of this work, different geographical clustering methods can be examined, as well as more contexts such as season or weather can be considered.

-----------------

The authors [119] proposed a new sequential prediction model according to the Markov chain model named SONG. According to this work, clients, are interested in visiting old POIs in the short term and tend to visit new POIs in the long run. This approach models the behavior and geographical impact of clients with a variable-order additive Markov chain. The Foursquare and JiePang [120] datasets are used to test this algorithm. Recall (Rec@k) and Normalized Discounted Cumulative Gain (NDCG@k) are employed to validate. Experimental results indicate that the proposed SONG substantially enhances the performance compared with the reference algorithms.

-----------------

A new POI recommendation model based on the Spatio-Temporal Activity Center POI (STACP) proposed by Rahmani et al. [127]; Considers the impact of a user's spatial and temporal characteristics jointly. This model, based on the matrix Factorization model, is statically trained according to the time feature and form centers of spatiotemporal activity for users and improves the quality of the recommendation. To evaluate the performance of this model, two popular datasets Foursquare and Gowalla, and also evaluation metrics Precision, Recall, and NDCG have been used. Experimental results demonstrate that the STACP method enhances statistical performance compared to state-of-the-art algorithms and illustrates the influence of utilizing geographic and temporal information in modeling client activity centers and the significance of their joint modeling. For higher improvement of this model, more information like comments of users and social relations can be added to the algorithm.

-----------------

The BPRN model was introduced by Hu et al., [133] who developed a new multi-layered neighbor-based BPR algorithm to investigate hidden information in recommending systems. The authors, based on analyzing the relationship between the item and the client, and examining several layers, determine that the item without ranking can be a desired and neighboring item for a user, and define the criteria for each layer. The items are then divided into different sets and arranged, and a personal, sorted list is specified for each user based on the model provided. Five datasets Movielens-100 k (ML-100 k), Movielens-1 m (ML-1 m), Ciao, Epinions, and Eachmovies [134] were used to test this algorithm, and also Precision, Recall, and F1 evaluation metrics were selected. The proposed method shows satisfactory results on the datasets. In the future, this approach to solving data sparsity and cold start issues could integrate multiple-layer analysis and also transfer learning.

  1. Section 4: this section just briefly describes some evaluation metrics and datasets. First, It is better to conclude the application scenarios of the different evaluation metrics. Second, it is better to provide the download link of the datasets.

RESPONSE:

Based on your valued and effective points, the evaluation metric part was reviewed and the required content was added as follows to this section. Also the download link for the datasets is provided. The data download link was also completed.

The purpose of this section is to review the evaluation metrics that most authors of recent state-of-the-arts have used to measure their POI recommender systems based on deep learning. These metrics have been observed at least once in selected studies. Most articles use accuracy, precision, recall, and other error-based metrics that are based on effectiveness and it indicates that recent investigations have prioritized accuracy. After them, the MAP evaluation metric has a more utilization in approaches and is used to evaluate object detection methods. According to Table 1, most LSTM-based studies have used this metric. F1 and NDCG are also very significant. F1 is suitable for evaluating binary classification systems and categorizes samples as positive and negative. NDCG metric is employed in recommender systems in various fields and determines how well the system is working. It is a popular way to measure the quality of a set of search outcomes and match the rating submitted by the user with the ideal rating. These iterative metrics can help to compare and replicate the results and can evaluate different research solutions.

  1. Section 5: thecontent of future work directions is not clear enough. It is better if the future research directions are listed for the soundness of readability. Besides, please describe the method of improving the model and future application scenarios in different paragraphs.

RESPONSE:

We are thankful to the reviewer for the thorough and constructive comments. Based on your guidance, the conclusion is improved as follows:

This is a systematic review aimed at reviewing common deep learning methods in POI recommendation systems and discussing the most prominent recent approaches in this field. In this research, several questions are presented and they are answered in different sections of the article. Recommender systems and influential factors and the challenges were expressed. Machine learning and deep learning models were introduced in this area and compare their advantages and disadvantages. The datasets and evaluation metrics used in recent approaches were categorized and the most important future directions and open problems were presented. The principal findings of this research can be outlined as follows:

  • Many recent approaches on POI recommendations have used combination methods with deep learning techniques to increase the accuracy of their recommendations and try to improve their suggestions to users;
  • Geographic data, the importance of similar friendships, user preferences, and temporal data are the most commonly used factors in modeling solutions related to POI recommendations;
  • The most widely used deep learning methods in POI recommendation are LSTM and CNN and the use of collaborative filtering techniques, matrix factorization, and its derivatives are popular along with other model;
  • Most of the datasets used in recent studies in the field of deep learning-based POI recommendation have been extracted from data related to well-known LBSNs such as Foursquare, Gowalla and Yelp Also, evaluation metrics for comparison between approaches are mostly based on Precision and Recall measures, which illustrates the attention to accuracy.

The most significant future directions and open issues in response to question 4 are summarized as follows:

  • To make progress towards more useful POI recommendation systems and to promote high quality research, authors should present their problem-oriented strategy with a combination of up-to-date methods according to the audience and their concerns and needs and offer a proper solution.
  • Researchers can develop a hybrid structure that incorporates the advantages of each technique and limits their disadvantages.
  • Developing models to improve the realization of user behavior by incorporating influential factors such as geographical, temporal, social factors, textual data, sentiment analysis of images and videos, upcoming events, weather, traffic, use of IoT components, and sensors can be effective and various methods like natural language processing (NLP) and topic modeling help to create novel POI recommendation techniques. Some of these extra features may be useful to the user, some may be more appealing to POI owners that depending on the type of issue, the best factors can be used.
  • Users may have different choices depending on their situation in the city or traveling. Restrictions on the user's choice of POI can be significant; Factors like a disease outbreak, telecommuting, the maximum travel length, the price of attractions, the unexpected crisis, staff strikes, and political conditions. It is recommended to use a more diverse dataset that examines the user from different angles, because the user may have checked-in under the conditions created by the system.
  • In addition to accuracy, other dimensions of quality like novelty and diversity can be considered in evaluations.
  • Recommending system security and users' privacy, paying attention to ethical issues, and preventing the misuse of users' information are also important problems that should be investigated, and providing personal advice and hiding users' sensitive information can be influential.
  • Another issue is that few studies have demonstrated the code and how to preprocess data and implement them, so publishing this content realistically could analyze the models and solutions better.

  1. It is better to compare some representative models. Such comparisons could be done by citing some state-of-the-art records reported by recent publications or conducting some experiments by the authors.

 RESPONSE:

We are thankful to the reviewer for the thorough and constructive comments. The desired items were added in the paper as follows:

We suggest a new pipeline named DeePOF [148] about POI recommendations and deep learning. The purpose of this technique is to gain the proper top-K point-of-interests sequence per customer. Our method utilizes the novel convolutional neural network and mean-shift clustering technique. We offer effective recommendations to users based on geographical and temporal information, as well as behavioral information from close friends. This hybrid technique is evaluated on two real datasets, Yelp and Gowalla of LBSNs. Six state-of-art approaches UFC [155], LFBCA[49], LORE[119], HGMAP[183], APOIR[198], and SAE-NAD [142] is selected for DeePOF performance validation and have been validated with two criteria Recall @ K and Precision @ K. The suggested prediction method results on Yelp and Gowalla datasets are represented in Figure7.

Figure 7. DeePOF [148] compared to 6 other models on the Gowalla and Yelp datasets.

  1. Literature review. Some related studies are missing. For example, latent factor analysis is a critical branch of recommendation. It is efficient in addressing the sparse issue of recommendation. It is better to discuss latent factor analysis-based recommender systems. There are many related papers, such as 1) A Data-Characteristic-Aware Latent Factor Model for Web Service QoS Prediction, IEEE Transactions on Knowledge and Data Engineering, vol. 34, no. 6, pp. 2525-2538, 2022. 2) An L1-and-L2-norm-oriented Latent Factor Model for Recommender Systems, IEEE Transactions on Neural Networks and Learning Systems, DOI: 10.1109/TNNLS.2021.3071392. 3) A posterior-neighborhood-regularized latent factor model for highly accurate web service QoS prediction, IEEE Transactions on Services Computing, vol. 15, no. 2, pp. 793-805, 1 March-April 2022. 4) A Deep Latent Factor Model for High-Dimensional and Sparse Matrices in Recommender Systems, IEEE Transactions on Systems Man and Cybernetics: Systems, vol. 51, no. 7, pp. 4285-4296, 2021, DOI:10.1109/TSMC.2019.2931393. 5) Di Wu, Peng Zhang, Yi He, and Xin Luo, A Double-Space and Double-Norm Ensembled Latent Factor Model for Highly Accurate Web Service QoS Prediction, IEEE Transactions on Services Computing,2022, doi: 10.1109/TSC.2022.3178543.

RESPONSE:

Based on your valued and effective points, the following content about latent factor analysis is added to the survey, and the mentioned references are cited:

Generally, there are two approaches to implementing collaborative filtering-based recommendation systems: nearest neighborhood analysis[96], [97] and latent factor analysis[98], [99]. In the first method, the recommender can compute the connection weight of the users or items to choose the nearest neighbors. The user's potential preferences are then predicted by examining the user's preferences for a new item based on last user data or highly related items. But the latent factor analysis-based method is a low-dimensional representation of items and users by which their dependencies can be accurately modeled. This method is derived from matrix factorization (MF)[100] approaches and it can discover low-rank feature matrices to define data and user ratings are factorized to an item and user feature vectors. Creates a series of loss functions based on known target matrix data that delays the desired latent factors, and then minimizes the resulting loss functions with respect to the desired latent factors to a technique with provides a low ranking with acceptable representation. This method focuses only on known data and is efficient in addressing the sparse issue of recommendation[101]–[104].

  1. Language issues, typos, and grammar errors.

RESPONSE:

We are grateful to you for your insightful and helpful comments. We have revised the paper according to the comments. We tried to edit the article's English language grammar and typos; we hope it will be accepted by you.

Once again, we would like to express our great appreciation to the editor and the reviewer for their careful readings and valuable comments on our paper. Looking forward to hearing from you.

Sincerely,

Sadaf Safavi, Mehrdad Jalali, Mahboobeh Houshmand

Reviewer 3 Report

This paper is in line with the scope of this journal with some significant contributions, but I am concerned about the following questions:

l  The authors should further analyze the results, In conclusion, I think the author's analysis of the research results is very shallow, so they must further summarize the research findings based on the research results

l   Please define all acronyms in the abstract. - State clearly the research question in the Introduction.

l   I think that not all symbols are defined for equations, especially in the part where a novel method is presented.

l  Expend conclusion to include details regarding the future work.

l   There are some technical and English language errors, please read the manuscript carefully and revise.

l  Please remove “we” from the manuscript and instead use the proposed system.

Author Response

Toward Point-of-Interest Recommendation Systems: a Critical Review on Deep Learning Approaches

Journal of electronics

Response to the Reviewers’ Comments

The reviewers’ high quality of refereeing helped us to clarify many areas in the paper. We greatly appreciate the editor-in-chief and reviewers again for giving us an opportunity to revise our manuscript. We incorporated the revisions recommended by the reviewers and carefully considered the reviewer comments which are attached. We provide below our responses and detailed revisions we have made to every comment that the reviewer has provided. We highlighted all revisions in the revised manuscript (Highlighted in yellow). Detailed descriptions are given in this revision note below.

Point-to-Point responses to the Reviewer.

Reviewer #3:

This paper is in line with the scope of this journal with some significant contributions, but I am concerned about the following questions:

  1. The authors should further analyze the results, In conclusion, I think the author's analysis of the research results is very shallow, so they must further summarize the research findings based on the research results.

RESPONSE:

We are thankful to the reviewer for the thorough and constructive comments. Based on your guidance, we summarize the research findings based on the research results as follows:

This is a systematic review aimed at reviewing common deep learning methods in POI recommendation systems and discussing the most prominent recent approaches in this field. In this research, several questions are presented and they are answered in different sections of the article. Recommender systems and influential factors and the challenges were expressed. Machine learning and deep learning models were introduced in this area and compare their advantages and disadvantages. The datasets and evaluation metrics used in recent approaches were categorized and the most important future directions and open problems were presented. The principal findings of this research can be outlined as follows:

  • Many recent approaches on POI recommendations have used combination methods with deep learning techniques to increase the accuracy of their recommendations and try to improve their suggestions to users;
  • Geographic data, the importance of similar friendships, user preferences, and temporal data are the most commonly used factors in modeling solutions related to POI recommendations;
  • The most widely used deep learning methods in POI recommendation are LSTM and CNN and the use of collaborative filtering techniques, matrix factorization, and its derivatives are popular along with other model;
  • Most of the datasets used in recent studies in the field of deep learning-based POI recommendation have been extracted from data related to well-known LBSNs such as Foursquare, Gowalla and Yelp Also, evaluation metrics for comparison between approaches are mostly based on Precision and Recall measures, which illustrates the attention to accuracy.
  •  
  1. Please define all acronyms in the abstract. - State clearly the research question in the Introduction.

RESPONSE:

Thank you for pointing to these items. All acronyms in the abstract are defined and the questions in the Introduction are corrected.

In recent years, location-based social networks (LBSNs) that allow members to share their location and provide related services, and point-of-interest (POIs) recommendations which suggest attractive places to visit have become noteworthy and useful for users, research areas, industries, and advertising companies. The POI recommendation system combines different information sources and creates numerous research challenges and questions. New research in this field utilizes deep learning techniques as a solution to the issues; because it has the ability to represent the non-linear relationship between users and items more effectively than other methods. Despite all the obvious improvements that have been made recently, this field still does not have an updated and integrated view of the types of methods, their limitations, features, and future prospects. This paper provides a systematic review focusing on recent research on this topic. First, this approach prepares an overall view of the types of recommendation methods, their challenges, and the various influencing factors that can improve model performance in POI recommendations, then reviews the traditional machine learning methods and deep learning techniques employed in the POI recommendation and analyzes their strengths and weaknesses. The recently proposed models are categorized according to the method used, the dataset, and the evaluation metrics. It found that these articles give priority to accuracy in comparison with other dimensions of quality. Finally, this approach introduces the research trends and future orientations and it realizes that POI recommender systems based on deep learning are a promising future work.

------------

In this work, our research questions are the following:

RQ1. What are the types of POI Recommender Systems, influencing factors, and challenges in their recommendation?

RQ2. What are the traditional machine learning methods and deep learning models that have been considered in recent POI recommender systems?

RQ3. What are the most widely employed evaluation metrics and popular datasets to evaluate POI recommendations based on deep learning?

RQ4. what are the most significant future research trends and open issues?

  1. I think that not all symbols are defined for equations, especially in the part where a novel method is presented.

RESPONSE:

Thank you for pointing to this item. We reviewed and defined all symbols in the article.

  1. Expend conclusion to include details regarding the future work.

RESPONSE:

Based on your valued and effective points, the following details regarding the future work were added to the conclusion:

The most significant future directions and open issues in response to question 4 are summarized as follows:

  • To make progress towards more useful POI recommendation systems and to promote high quality research, authors should present their problem-oriented strategy with a combination of up-to-date methods according to the audience and their concerns and needs and offer a proper solution.
  • Researchers can develop a hybrid structure that incorporates the advantages of each technique and limits their disadvantages.
  • Developing models to improve the realization of user behavior by incorporating influential factors such as geographical, temporal, social factors, textual data, sentiment analysis of images and videos, upcoming events, weather, traffic, use of IoT components, and sensors can be effective and various methods like natural language processing (NLP) and topic modeling help to create novel POI recommendation techniques. Some of these extra features may be useful to the user, some may be more appealing to POI owners that depending on the type of issue, the best factors can be used.
  • Users may have different choices depending on their situation in the city or traveling. Restrictions on the user's choice of POI can be significant; Factors like a disease outbreak, telecommuting, the maximum travel length, the price of attractions, the unexpected crisis, staff strikes, and political conditions. It is recommended to use a more diverse dataset that examines the user from different angles, because the user may have checked-in under the conditions created by the system.
  • In addition to accuracy, other dimensions of quality like novelty and diversity can be considered in evaluations.
  • Recommending system security and users' privacy, paying attention to ethical issues, and preventing the misuse of users' information are also important problems that should be investigated, and providing personal advice and hiding users' sensitive information can be influential.
  • Another issue is that few studies have demonstrated the code and how to preprocess data and implement them, so publishing this content realistically could analyze the models and solutions better.

  1. There are some technical and English language errors, please read the manuscript carefully and revise.

RESPONSE:

We are grateful to you for your insightful and helpful comments. We have revised the paper according to the comments. We tried to edit the article's English language grammar and style; we hope it will be accepted by you.

  1. Please remove “we” from the manuscript and instead use the proposed system.

RESPONSE:

Thank you for pointing to this item. We removed "we" from all contexts of the article and replaced words similar to the proposed system, for example:

This paper provides a systematic review focusing on recent research on this topic. First, this approach prepares an overall view of the types of recommendation methods, their challenges, and the various influencing factors that can improve model performance in POI recommendations, then reviews the traditional machine learning methods and deep learning techniques employed in the POI recommendation and analyzes their strengths and weaknesses. The recently proposed models are categorized according to the method used, the dataset, and the evaluation metrics. It found that these articles give priority to accuracy in comparison with other dimensions of quality. Finally, this approach introduces the research trends and future orientations and it realizes that POI recommender systems based on deep learning are a promising future work.

Once again, we would like to express our great appreciation to the editor and the reviewer for their careful readings and valuable comments on our paper. Looking forward to hearing from you.

Sincerely,

Sadaf Safavi, Mehrdad Jalali, Mahboobeh Houshmand

Round 2

Reviewer 2 Report

All my concerns have been addressed, this paper could be accepted.